# Quantum Oblivious Transfer: A Short Review

**DOI:** 10.3390/e24070945

**Published:** 2022-07-07

**Authors:** Manuel B. Santos, Paulo Mateus, Armando N. Pinto

**Affiliations:** 1Instituto de Telecomunicaçoes, 1049-001 Lisboa, Portugal; pmat@math.ist.utl.pt; 2Departamento de Matemática, Instituto Superior Técnico, Universidade de Lisboa, 1049-001 Lisboa, Portugal; 3Instituto de Telecomunicaçoes, 3810-193 Aveiro, Portugal; anp@ua.pt; 4Departamento de Eletrónica, Telecomunicaçoes e Informática, Universidade de Aveiro, 3810-193 Aveiro, Portugal

**Keywords:** quantum cryptography, oblivious transfer, secure multiparty computation, private database query

## Abstract

Quantum cryptography is the field of cryptography that explores the quantum properties of matter. Generally, it aims to develop primitives beyond the reach of classical cryptography and to improve existing classical implementations. Although much of the work in this field covers quantum key distribution (QKD), there have been some crucial steps towards the understanding and development of quantum oblivious transfer (QOT). One can show the similarity between the application structure of both QKD and QOT primitives. Just as QKD protocols allow quantum-safe communication, QOT protocols allow quantum-safe computation. However, the conditions under which QOT is fully quantum-safe have been subject to intense scrutiny and study. In this review article, we survey the work developed around the concept of oblivious transfer within theoretical quantum cryptography. We focus on some proposed protocols and their security requirements. We review the impossibility results that daunt this primitive and discuss several quantum security models under which it is possible to prove QOT security.

## 1. Introduction

Quantum technology has evolved to a point where it can be integrated into complex engineering systems. Most of the applications lie in the field of quantum cryptography, where one thrives to find protocols that offer some advantage over their classical counterparts. As analysed in [1,2], these advantages can be of two types:improve the security requirements, rendering protocols that are information-theoretically secure or require fewer computational assumptions;achieve new primitives that were previously not possible just with classical techniques.

Despite the most famous use-case of quantum cryptography being quantum key distribution (QKD), other primitives play an important role in this quest. Some examples of these cryptographic tasks are bit commitment [3], coin flipping [4], delegated quantum computation [5], oblivious transfer [6], position verification [7], and password-based identification [8,9].

The study of oblivious transfer (OT) has been very active since its first proposal in 1981 by Rabin [10] in the classical setting. Intriguingly enough, more than a decade earlier, a similar concept was proposed by Wiesner and rejected for publication due to the lack of acceptance in the research community. The importance of OT comes from its wide number of applications. More specifically, one can prove that OT is equivalent to the secure two-party computation of general functions [11,12], i.e., one can implement a secure two-party computation using OT as its building block. Additionally, this primitive can also be used for secure multi-party computation (SMC) [13], private information retrieval [14], private set intersection [15], and privacy-preserving location-based services [16]. More recently, the first direct quantum protocol for a generalization of oblivious transfer known as oblivious linear evaluation was proposed [17]. In addition, quantum versions of oblivious transfer have recently been applied to SMC systems in the field of genomics medicine [18,19]. Developing efficient and secure OT primitives is essential not only from a theoretical perspective but also from a practical one. When used in SMC systems, OTs mask every input bit [20] or mask the computation at every boolean circuit gate [21]. Regarding OT generation, a genomics use case [18,19] with three parties based on the Yao protocol [20] requires the execution of ∼12 million OTs. Hopefully, quantum OT approaches can be extended through classical methods [22,23], increasing the amount of OT produced. Since these OT extension methods only use symmetric cryptographic assumptions, they provide a better security level when compared with OT protocols based on asymmetric cryptography. Effectively, this hybrid approach is not attacked by Shor’s quantum algorithm [24]. Looking at the tree-party genomics use-case again, one only needs ∼25 thousand quantum base OTs when we use a classical OT extension protocol.

In a recent survey on classical OT [25], all the analysed protocols require some form of asymmetric cryptography. Indeed, in the classical setting, it is impossible to develop information-theoretic secure OT or even reduce it to one-way functions, requiring some public-key computational assumptions. As shown by Impaggliazzo and Rudich [26], one-way functions (symmetric cryptography) alone do not imply key agreement (asymmetric cryptography). Also, Gertner et al. [27] pointed out that since it is known that OT implies key agreement, this sets a separation between symmetric cryptography and OT, leading to the conclusion that OT cannot be generated alone by symmetric cryptography. This poses a threat to all classical OT protocols [28,29,30] that are based on mathematical assumptions provably broken by a quantum computer [24]. Besides the security problem, asymmetric cryptography tends to be more computationally complex than the symmetric one, creating a problem in terms of speed when a large number of OTs are required. Other approaches, usually named post-quantum, are still based on complexity problems and are not necessarily less complex, on the contrary, than the previously mentioned ones. The development of quantum OT tackles this issue, aiming to improve its security. Remarkably, there is a distinctive difference between classical and quantum OT from a security standpoint, as the latter is proved to be possible assuming only the existence of quantum-hard one-way functions [31,32]. This means quantum OT requires weaker security assumptions than classical OT.

Regarding efficiency, little work exists comparing classical and quantum approaches. This was recently initiated by Santos et al. [33], where the authors theoretically compared different classical OT approaches with the quantum BBCS (Bennett–Brassard–Crépeau–Skubiszewska) protocol in the $F_{\mathrm{com}}-$hybrid model (defined in Section 4.2). In addition, in subsequent work, Santos et al. [19] experimentally compared the efficiency impact of classical and quantum OT protocols on an SMC system.

In this paper, we review the particular topic of quantum oblivious transfer. We mainly comment on several important OT protocols, their underlying security models and assumptions, and how these contribute to the above points 1. and 2. in the quantum setting. To the best of our knowledge, there is no prior survey dedicated to quantum OT protocols alone. Usually, its analysis is integrated into more general surveys under the topic of “quantum cryptography”, leading to a less in-depth exposition of the topic. For reference, we provide some distinctive reviews on the general topic of quantum cryptography [1,34,35,36,37,38,39,40].

This review is divided as follows. In Section 2, we give some definitions of the primitives used throughout this work. Section 3 of this review contains a brief overview of the impossibility results related to OT. Section 4 provides an exposition of some of the most well-known quantum OT protocols based on some assumptions. Section 5 of this review is devoted to a relaxed version of the OT primitive. In Section 6, we review the work on a similar quantum primitive, private database query. Finally, we give a brief overview of topics not covered throughout this review (Section 7).

## 2. Definitions

For the sake of clarity, we present the definitions of the primitives used throughout this review.

**Definition** **1**(1-out-of-2 OT). *A 1-out-of-2 oblivious transfer is a two-party protocol between a sender S and a receiver R with the following specification:*
*The sender inputs two messages $m_{0},m_{1}\in {\left\{0,1\right\}}^{l}$ and outputs nothing.**The receiver inputs one bit choice $b\in \left\{0,1\right\}$ and outputs the corresponding message, i.e., $m_{b}$.*
*Moreover, it must satisfy the following security requirements:*

*Concealing: the sender knows nothing about the receiver bit choice b.*

*Oblivious: the receiver knows nothing about the message $m_{1-b}$.*



This definition can be generalized to the case of *k*-out-of-*N* OT, where the sender owns *N* messages, and the receiver can choose *k*. For k=1, this is commonly called private database query (PDQ). We may have different randomized versions of this primitive. We call *receiver random* 1-out-of-2 OT whenever the receiver’s bit choice is random; *sender random* 1-out-of-2 OT whenever the sender’s messages are random; *random* 1-out-of-2 OT whenever both input elements are random. We call “chosen” OT the non-randomized OT version of Definition 1.

**Definition** **2**(All-or-nothing OT). *An all-or-nothing oblivious transfer is a two-party protocol between a sender S and a receiver R with the following specification:*
*The sender inputs one message $m\in {\left\{0,1\right\}}^{l}$ and outputs nothing.**The receiver outputs with probability 1/2 the message m.*
*Moreover, it must satisfy the following security requirement:*

*Concealing: the sender does not know whether the receiver obtained her message or not.*



**Definition** **3**(Bit commitment). *A bit commitment is a two-phase reactive two-party protocol between a sender S, who wants to commit to some message m, and a receiver R:*
*Commitment phase: the sender inputs one message of the form (commit, m) and the receiver receives the confirmation that the sender has committed to some message.**Opening phase: the receiver asks the sender to open the commitment who reveals the message m to the receiver.*
*Moreover, it must satisfy the following security requirements:*

*Concealing (or hiding): the receiver knows nothing about the sender’s message m until the sender agrees to reveal it.*

*Binding: the sender is unable to change the message m after it is committed.*



## 3. Impossibility Results

The beginning of the development of quantum oblivious transfer (QOT) came hand in hand with the development of quantum bit commitment (QBC). In fact, the first proposed QOT protocol, known as the BBCS (Bennett–Brassard–Crépeau–Skubiszewska) protocol, reduces QOT to QBC [6]. This sets a distinctive difference between classical and quantum protocols. Although bit commitment (BC) can be reduced to oblivious transfer (OT) [12], the reverse is not true using only classical communication [41]. As pointed out by Salvail [41]: “classically, bit commitment can be built from any one-way function but oblivious transfer requires trapdoor one-way functions. It is very unlikely that one can find a proof that one-way functions and trapdoor one-way functions are in fact the same thing”. Therefore, Yao’s proof [42] of BBCS protocol [6] gives quantum communications the enhanced quality of having an equivalence between QOT and QBC—they can be reduced to each other—a relation that is not known and is very unlikely to exist in the classical realm.

At the time of the BBCS protocol, the quest for unconditionally secure QOT was based on the possibility of unconditional secure QBC. A year later, Brassard et al. presented a QBC protocol [43] named after the authors, BCJL (Brassard–Crépeau–Jozsa–Langlois). However, this work presented a flawed proof of its unconditional security which was generally accepted for some time, until Mayers spotted an issue on it [44]. Just one year after, Lo and Chau [45], and Mayers [46] independently proved unconditional QBC to be impossible. Nevertheless, the existence of unconditionally secure QOT not based on QBC was still put as an open question [34] even after the so-called no-go theorems [45,46]. However, Lo was able to prove directly that unconditionally secure QOT is also impossible [47]. He concluded this as a corollary of a more general result that states that secure two-party computations which allow only one of the parties to learn the result (one-side secure two-party computation) cannot be unconditionally secure. Lo’s results triggered a line of research on the possibility of two-sided secure two-party computation (both parties are allowed to learn the result without having access to the other party’s inputs), which was also proved by Colbeck to be impossible [48] and extended in subsequent works [49,50,51]. For a more in-depth review of the impossibility results presented by Lo, Chau and Mayers, we refer the interested reader to the following works [41,52].

Although the impossibility results have been well accepted in the quantum cryptography community, there was some criticism regarding the generality of the results [53,54,55,56]. This line of research reflects the view put forward by Yuen [53] in the first of these papers: “Since there is no known characterization of all possible QBC protocols, logically there can really be no general impossibility proof, strong or not, even if it were indeed impossible to have an unconditionally secure QBC protocol”. In parallel, subsequent analyses were carried out, reaffirming the general belief of impossibility [57,58,59]. However, most of the discord has ended with Ariano et al. proof [60] in 2007, giving an impossibility proof covering all conceivable protocols based on classical and quantum information theory. Subsequent work digested Ariano et al. [60] work, trying to present more succinct proofs [61,62,63] and to translate it into categorical quantum mechanics language [64,65,66].

Facing these impossibility results, the quantum cryptography community followed two main paths:Develop protocols under some assumptions (Section 4). These could be based on limiting the technological power of the adversary (e.g., noisy-storage model, relativistic protocols, isolated-qubit model) or assuming the existence of additional functionalities primitives (e.g., bit commitment).Develop protocols with a relaxed security definition of OT, allowing the adversary to extract, with a given probability, some information (partial or total) about the honest party input/output. This approach leads to the concepts of Weak OT (Section 5) and Weak Private Database Query (Section 6).

## 4. Qot Protocols with Assumptions

In this section, we explore protocols that circumvent the no-go theorems [45,46] utilizing some assumptions. Most of the presented solutions try to avoid using quantum-hard trapdoor one-way functions, making them fundamentally different from most post-quantum solutions that are based on trapdoor one-way functions. As an alternative, some of the presented solutions are based on one-way functions, which are believed to be quantum-hard [31,32,67], and others rely on some technological or physical limitation of the adversaries [68,69,70,71,72,73]. The latter are qualitatively different from complexity-based assumptions on which post-quantum protocols rely. In addition, all these assumptions have the important property that they only have to hold during the execution of the protocol for its security to be preserved. In other words, even if the assumptions lose their validity at some later point in time, the security of the protocol is not compromised, which also makes a major distinction from classical cryptographic approaches. This property is commonly known as *everlasting* security [74].

We start by presenting the first QOT protocol. We see how this leads to the development of two assumption models: $F_{\mathrm{COM}}-$hybrid model and the noisy-storage model. Then, we present the isolated-qubit model and how it leads to a QOT protocol. Finally, we review the possible types of QOT protocols under relativistic effects.

### 4.1. Bbcs Protocol

In 1983, Wiesner came up with the idea of *quantum conjugate coding* [75]. This technique is the main building block of many important quantum cryptographic protocols [8,76,77], including quantum oblivious transfer [6]. It also goes under the name of *quantum multiplexing* [77], *quantum coding* [78] or *BB84 coding* [41]. In quantum conjugate coding we encode classical information in two conjugate (non-orthogonal) bases. This allows us to have the distinctive property that measuring on one basis destroys the encoded information on the corresponding conjugate basis. So, when bit 0 and 1 are encoded by these two bases, no measurement is able to perfectly distinguish the states. Throughout this work, we will be using the following bases in the two-dimensional Hilbert space $H_{2}$:Computational basis: $+:=\{{|0\rangle }_{+},{|1\rangle }_{+}\}$;Hadamard basis: $\times :=\left\{{|0\rangle }_{\times },{|1\rangle }_{\times }\right\}=\left\{\frac{1}{\sqrt{2}}\left({|0\rangle }_{+}+{|1\rangle }_{+}\right),\frac{1}{\sqrt{2}}\left({|0\rangle }_{+}-{|1\rangle }_{+}\right)\right\}$.

**Protocol [6].** The first proposal of a quantum oblivious transfer protocol (BBCS protocol) is presented in Figure 1 and builds on top of the quantum conjugate coding technique. The sender S starts by using this coding to generate a set of qubits that are subsequently randomly measured by the receiver R. These two steps make up the first phase of the protocol that is also common to the BB84 QKD protocol. For this reason, it is called the *BB84 phase*. Next, with the output bits obtained by R and the random elements generated by S, both parties are ready to share a special type of key, known as *oblivious key*. This is achieved when S reveals her bases $\theta^{S}$ to R. Using the oblivious key as a resource, S can then obliviously send one of the messages $m_{0},m_{1}$ to R, ensuring that R is only able to know one of the messages. This is achieved using a two-universal family of hash functions F from ${\left\{0,1\right\}}^{n/2}$ to ${\left\{0,1\right\}}^{l}$. In addition, we use the notation $s\leftarrow_{\$}S$ to describe a situation where an element *s* is drawn uniformly at random from the set *S*.

**Oblivious keys.** The term *oblivious key* was used for the first time by Fehr and Schaffner [79] referring to a Random OT. However, under a subtle different concept, it was used by Jakobi et al. [80] as a way to implement Private Database Queries (PDQ), which we review in Section 6. In recent work, Lemus et al. [81] presented the concept of oblivious key applied to OT protocols. We can define it as follows.

**Definition** **4**(Oblivious key). *An oblivious key shared between two parties, sender S and receiver R, is a tuple $\mathrm{ok}:=\left({\mathrm{ok}}^{S},\left({\mathrm{ok}}^{R},e^{R}\right)\right)$ where ${\mathrm{ok}}^{S}$ is the sender’s key, ${\mathrm{ok}}^{R}$ is the receiver’s key and $e^{R}$ is the receiver’s signal string. $e^{R}$ indicates which indexes of ${\mathrm{ok}}^{S}$ and ${\mathrm{ok}}^{R}$ are correlated and which indexes are uncorrelated.*

The oblivious key ok shared between the two parties is independent of the sender’s messages $m_{0},m_{1}$ and is not the same as Random OT. As the sender S does not know the groups of indexes $I_{0}$ and $I_{1}$ deduced by R after the basis revelation, S does not have her messages fully defined. In addition, a similar concept was defined by König et al. [70] under the name of *weak string erasure*.

**Security.** Regarding security, the BBCS protocol is unconditionally secure against dishonest S. Intuitively, this comes from the fact that S does not receive any information from R other than some set of indexes $I_{0}$. However, the BBCS protocol is insecure against dishonest R. In its original paper [6], the authors describe a memory attack that provides R complete knowledge on both messages $m_{0}$ and $m_{1}$ without being detected. This can be achieved by having the receiver delay his measurements in step 2 to some moment after step 3. This procedure is commonly called the memory attack as it requires quantum *memory* to hold the states until step 3. The authors suggest that, for the protocol to be secure, the receiver has to be forced to measure the received states at step 2. In the following sections, we present two common approaches to tackle this issue. We may assume the existence of commitments or set physical assumptions that constrain R from delaying his measurement.

### 4.2. BBCS in the $F_{\mathrm{com}}$–Hybrid Model

**Model.** As mentioned in the previous section, a secure BBCS protocol requires the receiver R to measure his qubits in step 2. In this section, we follow the suggestion from the original BBCS paper [6] and fix this loophole using a commitment scheme. Since we assume we have access to some commitment scheme, we call it $F_{\mathrm{com}}$–hybrid model. The notation $F_{\mathrm{com}}$ is commonly used for ideal functionalities; however, here we abuse the notation by using $F_{\mathrm{com}}$ to refer to any commitment scheme (including the ideal commitment functionality).

**Protocol.** The modified BBCS (Figure 2) adds a *cut and choose* phase that makes use of a commitment scheme **com** to check whether R measured his qubits in step 2 or not. It goes as follows. R commits to the bases used to measure the qubits in the *BB84 phase* and the resulting output bits. Then, S chooses a subset of qubits to be tested and asks R to open the corresponding commitments of the bases and output elements. If no inconsistency is found, both parties can proceed with the protocol. Note that the size of the testing subset has to be proportional to *n* (security parameter), as this guarantees that the rest of the qubits were measured by R with overwhelming probability in *n*.

**Security.** Formally proving the security of this protocol led to a long line of research [6,9,31,32,42,79,82,83,84,85,86,87]. Earlier proofs from the 90s started by analyzing the security of the protocol against limited adversaries that were only able to do individual measurements [83]. Then, Yao [42] was able to prove its security against more general adversaries capable of doing fully coherent measurements. Although these initial works [42,83,84] were important to start developing a QOT security proof, they were based on unsatisfactory security definitions. At the time of these initial works, there was no composability framework [79,86] under which the security of the protocol could be considered. In modern quantum cryptography, these protocols are commonly proved in some quantum simulation-paradigm frameworks [9,70,79,86]. In this paradigm, the security is proved by showing that an adversary in a real execution of the protocol cannot cheat more than what he is allowed in an ideal execution, which is secure by definition. This is commonly proved by utilizing an entity, *simulator*, whose role is to guarantee that a real execution of the protocol is indistinguishable from an ideal execution. Moreover, they measured the adversary’s information through average-case measures (e.g., Collision Entropy, Mutual Information) which are proven to be weak security measures when applied to cryptography [88,89].

More desirable worst-case measures started to be applied to quantum oblivious transfer around a decade later [90,91]. These were based on the concept of *min-entropy* [88,89], $H_{\min}$, which, intuitively, reflects the maximum probability of an event to happen. More precisely, in order to prove security against dishonest receiver, one is interested in measuring the receiver’s min-entropy on the sender’s oblivious key ${\mathrm{ok}}^{S}$ conditioned on some quantum side information *E* he may has, i.e., $H_{\min}\left({\mathrm{ok}}^{S}|E\right)$. Informally, for a bipartite classical-quantum state $\rho_{XE}$ the conditional min-entropy $H_{\min}\left(X|E\right)$ is given by
$$H_{\min}{\left(X|E\right)}_{\rho_{XE}}:=-\log P_{guess}\left(X|E\right),$$
where $P_{guess}\left(X|E\right)$ is the probability the adversary guesses the value *x* maximized over all possible measurements. Damgård et al. [9] were able to prove the stand-alone QOT security when equipped with this min-entropy measure and with the quantum simulation-paradigm framework developed by Fehr and Schaffner [79]. Their argument to prove the protocol to be secure against dishonest receiver essentially works as follows. The cut and choose phase ensures that the receiver’s conditional min-entropy on the elements of ${\mathrm{ok}}^{S}$ belonging to $I_{1}$ (indexes with uncorrelated elements between S and R oblivious keys) is lower-bounded by some value that is proportional to the security parameter, i.e., $H_{\min}\left({\mathrm{ok}}_{I_{1}}^{S}|E\right)\geq n\lambda$ for some λ>0. Note that this is equivalent to derive an upper bound on the guessing probability $P_{guess}\left({\mathrm{ok}}_{I_{1}}^{S}|E\right)\leq 2^{-n\lambda }$. Having deduced an expression for λ, they proceed by applying a random hash function *f* from a two-universal family F, $f\leftarrow_{\$}F$. This final step ensures that $f({\mathrm{ok}}_{I_{1}}^{S})$ is statistically indistinguishable from uniform (privacy amplification theorem [90,91,92]). The proof provided by Damgård et al. [9] was extended by Unruh [86] to the quantum Universal Composable model, making use of ideal commitments. Now, a natural question arises: which commitment schemes can be used to render simulation-based security?

**Commitment scheme.** The work by Aaronson [67] presented a non-constructive proof that “indicates that collision-resistant hashing might still be possible in a quantum setting”, giving confidence in the use of commitment schemes based on quantum-hard one-way functions in the $\Pi_{F_{\mathrm{com}}}^{\mathrm{BBCS}}$ protocol. Hopefully, it was shown that commitment schemes can be built from any one-way function [93,94,95], including quantum-hard one-way functions. Although it is intuitive to plug in into $\Pi_{F_{\mathrm{com}}}^{\mathrm{BBCS}}$ a commitment scheme derived from a quantum-hard one-way function, this does not necessarily render a simulation-based secure protocol. This happens because the nature of the commitment scheme can make the simulation-based proof difficult or even impossible. For a detailed discussion see [31].

Indeed, the commitment scheme must be quantum secure. In addition, the simulator must have access to two intriguing properties: *extractability* and *equivocality*. Extractability means the simulator can extract the committed value from a malicious committer. Equivocal means the simulator can change the value of a committed value at a later time. Although it seems counter-intuitive to use a commitment scheme where we can violate both security properties (hiding and biding properties), it is fundamental to prove its security. Extractability is used by the simulator to prove security against the dishonest sender and equivocality is used by the simulator to prove security against the dishonest receiver. In the literature, there have been some proposals of the commitment schemes $F_{\mathrm{com}}$ with these properties based on:Quantum-hard one-way functions [31,32];Common Reference String (CRS) model [86,96];Bounded-quantum-storage model [97];Quantum hardness of the Learning With Errors assumption [9].

**Composability.** The integration of secure oblivious transfer executions in secure-multiparty protocols [11] should not lead to security breaches. Although it seems intuitive to assume that a secure OT protocol can be integrated within more complex protocols, proving this is highly non-trivial as it is not clear a priori under which circumstances protocols can be composed [98].

The first step towards composability properties was the development of simulation based-security, however, this does not necessarily imply composability (see Section 4.2 of [98] for more details). A *composability framework* is also required. In the literature, there have been some proposals for such a framework. In summary, Fehr and Schaffner [79] developed a composability framework that allows sequential composition of quantum protocols in a classical environment. The works developed by Ben-Or and Mayers [99] and Unruh [86,100] extended the classical Universal Composability model [101] to a quantum setting (quantum-UC model), allowing concurrent composability. Maurer and Renner [102] developed a more general composability framework that does not depend on the models of computation, communication, and adversary behaviour. More recently, Broadbent and Karvonen [66] created an abstract model of composable security in terms of category theory. Up until now, and to the best of our knowledge, the composable security of the protocol $\Pi_{F_{\mathrm{com}}}^{\mathrm{BBCS}}$ was only proven in the Fehr and Schaffner model [79] by Damgård et al. [9] and in the quantum-UC by Unruh [86].

### 4.3. BBCS in the Limited-Quantum-Storage Model

In this section, we review protocols based on the limited-quantum-storage model. The protocols developed under this model avoid the no-go theorems because they rely their security on reasonable assumptions regarding the storage capabilities of both parties. Under this model, there are mainly two research lines. One was started by Damgård, Fehr, Salvail and Schaffner [68], who developed the bounded-storage model. In this model, the parties can only store a limited number of qubits. The other research line was initiated by Wehner, Schaffner and Terhal [69], who developed the noisy-storage model. In this model the parties can store *all* qubits. However, they are assumed to be unstable, i.e., they only have imperfect noisy storage of qubits that forces some decoherence. In both models, the adversaries are forced to use their quantum memories as both parties have to wait a predetermined time (Δt) during the protocol.

#### 4.3.1. Bounded-Quantum-Storage Model

**Model.** In the bounded-quantum-storage model or BQS model for short, we assume that, during the waiting time Δt, the adversaries are only able to store a fraction 0<γ<1 of the transmitted qubits, i.e., the adversary is only able to keep q=nγ qubits. The parameter γ is commonly called the storage rate.

**Protocol.** The protocol in the BQS model, $\Pi_{\mathrm{bqs}}^{\mathrm{BBCS}}$, is very similar to the BBCS protocol $\Pi^{\mathrm{BBCS}}$ presented in Figure 1. The difference is that both parties have to wait a predetermined time (Δt) after step 2. This protocol is presented in Figure 3.

**Security.** We just comment on the security against a dishonest receiver because the justification for the security against a dishonest sender is the same as in the original BBCS protocol, $\Pi^{\mathrm{BBCS}}$ (see Section 4.1).

Under the BQS assumption, the waiting time (Δt) effectively prevents the receiver from holding *a large fraction* of qubits until the sender reveals the bases choices $\theta^{S}$ used during the *BB84 phase*. This comes from the fact that a dishonest receiver is forced to measure a fraction of the qubits, leading him to lose information about the sender’s bases $\theta^{S}$.

More specifically, Damgård et al. [91] showed that, with overwhelming probability, the loss of information about the sender’s oblivious key (${\mathrm{ok}}_{I_{1}}^{S}$) is described by a lower bound on the min-entropy [37]
$$H_{\min}\left({\mathrm{ok}}_{I_{1}}^{S}|E\right)\geq \frac{1}{4}n-\gamma n-l-1.$$
Similarly to the $F_{\mathrm{com}}-$hybrid model, the min-entropy value has to be bounded by a factor proportional to the security parameter *n*. To render a positive bound, we derive an upper bound on the fraction of qubits that can be saved in the receiver’s quantum memory, while preserving the security of the protocol, i.e., $\gamma <\frac{1}{4}$.

The above upper bound was later improved by König et al. [70] to $\gamma <\frac{1}{2}$. The authors also showed that the BQS model is a special case of the noisy-quantum-storage model. Subsequently, based on higher-dimensional mutually unbiased bases, Mandayam and Wehner [103] presented a protocol that is still secure when an adversary cannot store even a small fraction of the transmitted pulses. In this latter work, the storage rate γ approaches 1 for increasing dimension.

**Composability.** The initial proofs given by Damgård et al. [68,91] were only developed under the stand-alone security model [104]. In this model the composability of the protocol is not guaranteed to be secure. These proofs were extended by Wehner and Wullschleger [104] to a simulation-based framework that guarantees sequential composition. In addition, in a parallel work, Fehr and Schaffner developed a sequential composability framework under which $\Pi_{\mathrm{bqs}}^{\mathrm{BBCS}}$ is secure considering the BQS model.

The more desirable quantum-UC framework was extended by Unruh and combined with the BQS model [97]. In Unruh’s work, he developed the concept of BQS-UC security which, as in UC security, implies a very similar composition theorem. The only difference is that in the BQS-UC framework we have to keep track of the quantum memory-bound used by the machines activated during the protocol. Under this framework, Unruh follows a different approach as he does not use the protocol $\Pi_{\mathrm{bqs}}^{\mathrm{BBCS}}$ (Figure 3). He presents a BQS-UC secure commitment protocol and composes it with the $\Pi_{F_{\mathrm{com}}}^{\mathrm{BBCS}}$ protocol (Figure 2) in order to get a constant-round protocol that BQS-UC-emulates any two-party functionality.

#### 4.3.2. Noisy-Quantum-Storage Model

**Model.** The noisy-quantum-storage model, or NQS model for short, is a generalization of the BQS model. In the NQS model, the adversaries are allowed to keep any fraction ν of the transmitted qubits (including the case ν=1) but their quantum memory is assumed to be noisy [70], i.e., it is impossible to store qubits for some amount of time (Δt) without undergoing decoherence.

More formally, the decoherence process of the qubits in the noisy storage is described by a completely positive trace preserving (CPTP) map (also called channel) $F:B\left(H_{\mathrm{in}}\right)\to B\left(H_{\mathrm{out}}\right)$, where $H_{\mathrm{in}/\mathrm{out}}$ is the Hilbert space of the stored qubits before (in) and after (out) the storing period Δt and B(H) is the set of positive semi-definite operators with unitary trace acting on an Hilbert space H. F receives a quantum state $\rho \in H_{\mathrm{in}}$ at time *t* and outputs a quantum state $\rho^{\prime }\in H_{\mathrm{out}}$ at a later time t+Δt.

With this formulation, we can easily see that the BQS model is a particular case of the NQS. In BQS, the channel is of the form $F={𝟙}^{⊗\nu n}$, where the storage rate ν is the fraction of transmitted qubits stored in the quantum memory. The most studied scenario is restricted to n−fold quantum channels, i.e., $F=N^{⊗\nu n}$ [69,70,105], where the channel N is applied independently to each individual stored qubit. In this particular case, it is possible to derive specific security parameters.

**Protocols.** The protocol from BQS model $\Pi_{\mathrm{bqs}}^{\mathrm{BBCS}}$ is also considered to be secure in the NQS model [105]. However, the first proposed protocol analysed in this general NQS model was developed by König et al. [70]. This protocol draws inspiration from the research line initiated by Cachin, Crépeau and Marcil [106] about classical OT in the bounded-classical-storage model [107,108]. Similar to these works [106,107,108], the protocol presented by König et al. [70] uses the following two important techniques in its classical post-processing phase: encoding of sets and interactive hashing. The former is defined as an injective function $\mathrm{Enc}:{\left\{0,1\right\}}^{t}\to T$, where *T* is a set of all subsets of [n] with size n/4. The latter is a two-party protocol between Alice and Bob with the following specifications. Bob inputs some message $W^{t}$ and both parties receive two messages $W_{0}^{t}$ and $W_{1}^{t}$ such that there exists some b∈{0,1} with $W_{b}^{t}=W^{t}$. The index *b* is unknown to Alice, and Bob has little control over the choice of the other message $W^{t}$, i.e., it is randomly chosen by the functionality.

In this review, we only present the naïve protocol presented in the original paper [70] as it is enough to give an intuition on the protocol. Although both $\Pi_{\mathrm{bqs}}^{\mathrm{BBCS}}$ and $\Pi_{\mathrm{nqs}}^{\mathrm{BBCS}}$ protocols are different, we keep a similar notation for a comparison purpose. The protocol $\Pi_{\mathrm{nqs}}^{\mathrm{BBCS}}$ (Figure 4) goes as follows. The first two phases (*BB84* and *Waiting time*) are the same as in $\Pi_{\mathrm{bqs}}^{\mathrm{BBCS}}$ (Figure 3). Then, both parties generate a very similar resource to oblivious keys, named *weak string erasure* (WSE). After this WSE process, the sender also holds the totality of the key ${\mathrm{ok}}^{S}$, while the receiver holds a fourth of this key, i.e., the tuple $\left(I,{\mathrm{ok}}^{R}:={\mathrm{ok}}_{I}^{S}\right)$ where *I* is the set of indexes they measured in the same basis and its size is given by $\left|I\right|=\frac{n}{4}$. Then, along with a method of encoding sets into binary strings, both parties use interactive hashing to generate two index subsets, $I_{0}$ and $I_{1}$, where the sender plays the role of Alice, and the receiver plays the role of Bob. The two subsets ($I_{0}$ and $I_{1}$) together with two 2–universal hash functions are enough for the sender to generate her output messages $\left(m_{0},m_{1}\right)$ and the receiver to get his bit choice along with the corresponding message $\left(b,m_{b}\right)$. For more details on the protocols for encodings of sets and interactive hashing, we refer to Ding et al. [107] and Savvides [108].

**Security.** Based on the original BQS protocol (Figure 3), the first proofs in the NQS model were developed by Schaffner, Wehner and Terhal [69,109]. However, in these initial works, the authors only considered individual-storage attacks, where the adversary treats all incoming qubits equally. Subsequently, Schaffner [105] was able to prove the security of $\Pi_{\mathrm{bqs}}^{\mathrm{BBCS}}$ against arbitrary attacks in the more general NQS model defined by König et al. [70].

In this more general NQS model, the security of both protocols $\Pi_{\mathrm{bqs}}^{\mathrm{BBCS}}$ and $\Pi_{\mathrm{nqs}}^{\mathrm{BBCS}}$ (Figure 3 and Figure 4) against a dishonest receiver depends on the possibility to set a lower-bound on the min-entropy of the “unknown” key ${\mathrm{ok}}_{I_{1-b}}^{S}$ given the receiver’s quantum side information. His quantum side information is given by the output of the quantum channel F when applied to the received states. More formally, one has to lower-bound the expression $H_{\min}\left({\mathrm{ok}}_{I_{1-b}}^{S}|F\left(Q_{\mathrm{in}}\right)\right)$, where $Q_{\mathrm{in}}$ denotes the subsystem of the received states before undergoing decoherence. It is proven [70] that this lower-bound depends on the receiver’s maximal success probability of correctly decoding a randomly chosen n-bit string $x\in {\left\{0,1\right\}}^{n}$ sent over the quantum channel F, i.e., $P_{\mathrm{succ}}^{F}\left(n\right)$. This result is given by Lemma 1.

**Lemma** **1**(Lemma II.2. from [70]). *Consider an arbitrary ccq-state $\rho_{XTQ}$, and let $\varepsilon ,\varepsilon^{\prime }>0$ be arbitrary. Let $F:B\left(H_{Q_{\mathrm{in}}}\right)\to B\left(H_{Q_{\mathrm{out}}}\right)$ be an arbitrary CPTP map, where $H_{Q_{\mathrm{in}}}$ and $H_{Q_{\mathrm{out}}}$ are the Hilbert space corresponding to the subsystem $Q_{\mathrm{in}}$ and $Q_{\mathrm{out}}$, respectively. Then,*
$$H_{\min}^{\varepsilon +\varepsilon^{\prime }}\left(X|TF\left(Q\right)\right)\geq -\log P_{\mathrm{succ}}^{F}\left(\left\lfloor H_{\min}^{\varepsilon }\left(X|T\right)-\log\frac{1}{\varepsilon }\right\rfloor \right),$$*where $H^{\epsilon }$ denotes the smooth min-entropy.*

For particular channels $F=N^{⊗\nu }$, König et al. [70] concluded that security in the NQS model can be obtained in case
$$C_{N}\cdot \nu <\frac{1}{2},$$
where $C_{N}$ is the classical capacity of quantum channels N satisfying a particular property (strong-converse property).

### 4.4. Device-Independent QOT in the Limited-Quantum-Storage Model

In addition to the presented assumptions (e.g., existence of a commitment scheme or limited-quantum-storage model), the corresponding protocols also assume that dishonest parties cannot corrupt the devices of honest parties. In other words, the protocols’ security depends on the guarantee given by the parties that their quantum devices behave as specified during the protocol execution. However, quantum hacking techniques pose a security threat to these protocols. Santos et al. [19] gave a brief description of how two common techniques (faked-state and trojan-horses attacks) break the security of assumption-based BBCS protocols ($\Pi_{F_{\mathrm{com}}}^{\mathrm{BBCS}}$, $\Pi_{\mathrm{bqs}}^{\mathrm{BBCS}}$ and $\Pi_{\mathrm{nqs}}^{\mathrm{BBCS}}$). In summary, a faked-state attack allows the receiver to avoid the security guarantees enforced by the assumptions and effectively receive both messages $m_{0}$ and $m_{1}$. More shockingly, both attacks allow the sender to find the receiver’s bit choice *b*, which is proved to be *unconditionally* secure with trusted devices. Nevertheless, to the best of our knowledge, a more detailed study on the effects of quantum hacking techniques on QOT protocols is lacking in the literature. For a more in-depth review of quantum hacking techniques applied to QKD systems, we refer to Sun and Huang [40] and Pirandola et al. [38].

There is a research line focused on the study of security patches for each technological loophole [110]. However, this approach pursues the difficult task of approximating the experimental implementations to the ideal protocols. It would be more desirable to develop protocols that already consider faulty devices and are robust against any kind of quantum hacking attack. This is the main goal of device-independent (DI) cryptography, where we drop the assumption that quantum devices cannot be controlled by the adversary and we treat them simply as black-boxes [111,112]. In this section, we give a general overview of the state-of-the-art of DI protocols. For a more in-depth description, we refer to the corresponding original works.

**Kaniewski-Wehner DI protocol [113].** The first DI protocol of QOT was presented in a joint work by Kaniewski and Wehner [113] and its security proof was improved by Ribeiro et al. [114]. The protocol was proved to be secure in the noisy-quantum-storage (NQS) model as it uses the original NQS protocol $\Pi_{\mathrm{nqs}}^{\mathrm{BBCS}}$ (Figure 4) for trusted devices. It analyzes two cases leading to slightly different protocols.

First, they assume that the devices behave similarly every time they are used (*memoryless assumption*). This assumption allows for testing the devices independently from the actual protocol, leading to a DI protocol in two phases: *device-testing phase* and *protocol phase*. Under this memoryless assumption, one can prove that the protocol is secure against general attacks using proof techniques borrowed from [70]. Then, they analyse the case *without* the memoryless assumption. In that case it is useless to test the devices in advance as they can change their behaviour later. Consequently, the structure of the initial DI protocol (with two well-separated phases) has to be changed to accommodate this more realistic scenario. That is, the rounds for the device-testing phase have to be intertwined with the rounds for the protocol phase.

As a common practice in DI protocols, the DI property comes from some violation of Bell inequalities [115], which ensures a certain level of entanglement. This means that, in the protocol phase, the entanglement-based variant of $\Pi_{\mathrm{nqs}}^{\mathrm{BBCS}}$ must be used. Here, the difference lies in the initial states prepared by the sender, which, for this case, are maximally entangled states $\left|\Phi^{+}\rangle \langle \Phi^{+}\right|$ where $|\Phi^{+}\rangle =\frac{1}{\sqrt{2}}\left(|00\rangle +|11\rangle \right)$. The Bell inequality used in this case comes from the Clauser-Holt-Shimony-Horne (CHSH) inequality [116].

**Broadbent-Yuen DI protocol [117].** More recently, Broadbent and Yuen [117] used the $\Pi_{\mathrm{bqs}}^{\mathrm{BBCS}}$ (Figure 3) to develop a DI protocol in the BQS model. Similar to Kaniewski and Wehner’s work, they the protocol to be secure under the memoryless assumption. However, they do not require non-communication assumptions that ensure security from Bell inequality violations. Instead of using the CSHS inequality, their work is based on a recent self-testing protocol [118,119] based on a post-quantum computational assumption (hardness of Learning with Errors (LWE) problem [120]).

**Ribeiro-Wehner MDI protocol [121].** Ribeiro and Wehner [121] developed an OT protocol in the measurement-device-independent (MDI) regime [122] to avoid the technological challenges in the implementation of DI protocols [123]. In addition, this work was motivated by the fact that, so far, there is no security proof in the DI setting. Furthermore, many attacks on the non device-independent protocols affect the measurement devices rather than the sources [124]. The presented protocol follows the research line of König et al. [70] and start by executing a weak string erasure in the MDI setting (*MDI-WSE phase*). For this reason, it is also proved to be secure in the NQS model.

The initial MDI-WSE phase goes as follows. Both the sender and receiver send random states ${|x^{S}\rangle }_{\theta^{S}}$ and ${|x^{R}\rangle }_{\theta^{R}}$, respectively, to an external agent that can be controlled by the dishonest party. The external agent performs a Bell measurement on both received states and announces the result. The receiver flips his bit according to the announced result to match the sender’s bits. Then, both parties follow the $\Pi_{\mathrm{nqs}}^{\mathrm{BBCS}}$ protocol (Figure 4) from the waiting time phase onward. A similar protocol was presented by Zhou et al. [125] which additionally takes into account error estimation to improve the security of the protocol.

### 4.5. Otm in the Isolated-Qubits Model

**One-Time Memory.** A One-Time Memory (OTM) is a cryptographic device that allows more generic functionalities such as One-Time Programs [126]. Its definition is similar to 1-out-of-2 Oblivious Transfer: the sender writes two messages $m_{0}$ and $m_{1}$ into the OTM and sends the OTM to the receiver. The receiver can then run the OTM only once and choose one of the messages, $m_{b}$, while staying oblivious about the other message, $m_{1-b}$. The main difference between OT and OTM is that in OT the sender learns whether the receiver has received the message $m_{b}$, while in OTM, the sender does not receive any confirmation about that. This difference comes from the identifying feature of one-way communication in OTM [39]: after the sender handles the OTM device to the receiver, there is no more communication between the parties.

**Model.** In the isolated-qubits model, we assume that qubits cannot be entangled and can only be handled through single-qubit measurements. More specifically, this model only allows dishonest parties to perform local operations and classical communication while preparing the OTM device (sender) or reading it (receiver). As Liu [71] comments in his original article about quantum-based OTM, the isolated-qubits model complements the limited-quantum-storage models. Indeed, the isolated-qubits model does not allow the parties to perform entanglement and assumes the existence of long-term memories. On the other hand, the limited-quantum-storage models allow the existence of entanglement but assume qubits cannot be stored for a long time.

**Protocol [71].** Liu presented the first protocol [71] for quantum OTM, which is also based on the standard idea of conjugate coding. In this protocol, the sender uses the computational and hadamard bases to prepare the states (grey lines in Figure 5) and the receiver uses the bases $B_{0}=\left\{|\beta_{\frac{\pi }{8}}\rangle ,|\beta_{\frac{5\pi }{8}}\rangle \right\}$ and $B_{1}=\left\{|\beta_{-\frac{\pi }{8}}\rangle ,|\beta_{\frac{3\pi }{8}}\rangle \right\}$ to measure the received qubits (red lines in Figure 5).

**Figure 5 entropy-24-00945-f005:**
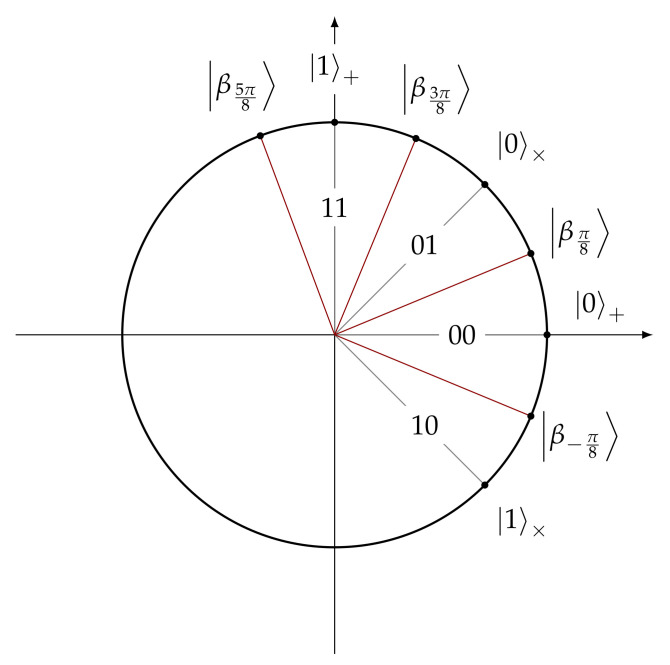
Quantum states used in the $\Pi_{\mathrm{iq}}^{\mathrm{OTM}}$ protocol.

So, the protocol goes as follows. The sender prepares a string of isolated qubits, $|\alpha_{a_{i}b_{i}}\rangle$ for i∈[n], using the computational and hadamard bases according to the following encoding:$$\begin{array}{ccc} |\alpha_{00}\rangle & = & {|0\rangle }_{+}\\ |\alpha_{11}\rangle & = & {|1\rangle }_{+}\\ |\alpha_{01}\rangle & = & {|0\rangle }_{\times }\\ |\alpha_{10}\rangle & = & {|1\rangle }_{\times }. \end{array}$$
The choice of $a_{i}$ and $b_{i}$ in $\alpha_{a_{i}b_{i}}$ depends on the sender’s messages $\left(m_{0},m_{1}\right)$ and two random functions set as protocol parameters $C,D:{\left\{0,1\right\}}^{l}\to {\left\{0,1\right\}}^{n}$, which, with high probability, are good error correcting codes. More specifically,
$$\begin{array}{ccc} a_{i} & = & C{\left(m_{0}\right)}_{i}\\ b_{i} & = & D{\left(m_{1}\right)}_{i}. \end{array}$$

The intuition behind the correctness of the protocol is that this qubit encoding allows the receiver to get a noisy version of either $C(m_{0})$ or $D(m_{1})$ when he uses basis $B_{0}$ or $B_{1}$ to measure all qubits, respectively. We can check this is the case based on Figure 5. Consider the case where the receiver chooses to read message b=0. This case means he will measure all the qubits in the $B_{0}$ basis. Imagine the receiver obtains the state $|\beta_{\frac{\pi }{8}}\rangle$ after measuring the i-th qubit. Consequently, the receiver will set $C{\left(m_{0}\right)}_{i}=0$, since, with higher probability, the initial qubit state was prepared in one of the adjacent vectors, i.e., ${|0\rangle }_{\times }$ (encoding 01) or ${|0\rangle }_{+}$ (encoding 00). However, this guess may came with some error, as the states ${|1\rangle }_{\times }$ and ${|1\rangle }_{+}$ are not orthogonal to the obtained state $|\beta_{\frac{\pi }{8}}\rangle$. The protocol is described in Figure 6.

**Security.** The LOCC assumption (local operations and classical communication) is crucial to ensure the security of the protocol because there is a joint measurement that allows recovering both messages $m_{0}$ and $m_{1}$. In the original paper [71], Liu proved that the state prepared by the sender can be distinguished almost perfectly by a measurement that uses entanglement among the *n* qubits. This distinguishability is achieved using the common technique of “pretty good measurement” [127].

The security proof of the $\Pi_{\mathrm{iq}}^{\mathrm{OTM}}$ protocol is presented with some caveats that fostered some subsequent work [128,129]. Most importantly, the adversary can obtain partial knowledge of both messages as it is not clear how the parties can engage in a privacy amplification phase without communication. This led to the definition of a weaker notion of OTM where the possibility of having partial knowledge of both messages was included. Intuitively, the definition states that a *leaky* OTM is an OTM with the additional property of having min-entropy of both messages $m_{0}$ and $m_{1}$ approximately lower-bounded by the length of one message, *l*, i.e., $H_{\min}\left(m_{0},m_{1}|E\right)\geq \left(1-\delta \right)l$ for δ>0.

**Further work.** In the original paper, [71], the leaky security of $\Pi_{\mathrm{iq}}^{\mathrm{OTM}}$ was only proved using a weaker entropy measure (Shannon entropy) and assuming only one-pass LOCC adversaries, i.e., adversaries that can only measure each qubit once. Subsequently, Liu [128] was able to improve on the previous work and proved a modified version of $\Pi_{\mathrm{iq}}^{\mathrm{OTM}}$ to be a leaky OTM, which is stated secure in terms in terms of the (smoothed) min-entropy. Finally, Liu [129] proposed a variant of privacy amplification which uses a *fixed* hash function F. This allows to building a protocol for (not leaky) single-bit OTM that is secure in the isolated qubits model.

### 4.6. Qot in a Relativistic Setting

In this section, we present two variants of oblivious transfer that take into account special relativity theory. These two variants do not exactly follow the OT definition as it was proved that it is impossible to construct unconditionally secure OT even under the constraints imposed by special relativity [48,130,131,132,133].

**Model.** In the relativistic setting, we consider protocols that take into account the causality of Minkowski space-time, limiting the maximum possible signalling speed (no-superluminal principle) [72].

#### 4.6.1. Spacetime-Constrained Oblivious Transfer

The cryptographic task of spacetime-constrained oblivious transfer (SCOT) is motivated by the following scenario. The sender has two computers $C_{0}$ at x=−h and $C_{1}$ at x=h, which can only be accessed within regions of space-time denoted by $R_{0}$ and $R_{1}$ using passwords $m_{0}$ and $m_{1}$, respectively (Figure 7). This setup can be applied to spacetime-constrained multiparty computation [72].

**Definition.** In SCOT, the sender inputs two messages $m_{0}$ and $m_{1}$ and the receiver one-bit choice *b*. The receiver obtains message $m_{b}$ within some space-time region $R_{b}$ (Figure 7) and the sender stays oblivious about his bit choice *b*. Furthermore, the receiver is not able to know anything about the other message $m_{1-b}$ at space-time region $R_{1-b}$. The fundamental difference between the standard 1-out-of-2 OT and SCOT is related to the space-time regions in which the receiver is allowed to know messages $m_{0}$ and $m_{1}$. In the standard OT, the receiver can never deduce the message $m_{1-b}$, whether in SCOT the receiver is allowed to find the message $m_{1-b}$ outside region $R_{1-b}$.

**Protocol [72].** In the first proposed SCOT protocol [72], both the sender and receiver have three representatives (called agents) who take part in the protocol at different spacetime locations. The sender’s agents are denoted by $S_{0}$, S and $S_{1}$ and the receiver’s agents by $R_{0}$, R and $R_{1}$, which are located at x=−h, x=0 and x=h, respectively (Figure 7). The protocol is also based on the standard idea of conjugate coding and it goes as follows. The agent S prepares a string of qubits using conjugate coding and sends them to the receiver’s corresponding agent R at spacetime point *P*. Then, S sends the bases θ used to prepare these states and masked messages $t_{i}$ to the agents $S_{i}$ at $Q_{i}$, for i=0,1 (blue arrows in Figure 7). Then, the receiver’s agent R sends the received qubits ${|x\rangle }_{\theta }$ to the agent $R_{b}$ located at $Q_{b}$ according to his bit choice *b*. In Figure 7, it is depicted the case where the receiver’s bit choice is b=1, meaning R sends the string of quibits to $R_{1}$ (yellow arrow) at $Q_{1}$. Upon receiving the tuple $\left(\theta ,t_{i}\right)$, the agent $S_{i}$ sends them to the corresponding receiver’s agent $R_{i}$. At this stage, $R_{b}$ has all the necessary elements to decode $t_{b}$ and retrieve the desired message $m_{b}$. Check the protocol in Figure 8 for more details.

**Security.** Regarding security, the general no-go theorems do not apply to this SCOT protocol because of the Minkowski causality. The causality implies that any nonlocal unitary applied within both spacetime regions $R_{0}$ and $R_{1}$, can only be completed in the future light cone of point *Q*. In other words, the attack cannot be achieved within both spacetime regions $R_{0}$ and $R_{1}$.

**Further work.** The protocol $\Pi^{\mathrm{SCOT}}$ was improved in a subsequent work [134], allowing more practical implementation of SCOT. This improved protocol does not require quantum memories and long-distance quantum communications. Then, the protocol presented by Garcia and Kerenidis [134] was extended to one-out-of-*k* SCOT, where the sender owns *k* messages and the receiver gets just one of the messages without letting the sender know his choice [135].

#### 4.6.2. Location-Oblivious Data Transfer

Location-oblivious data transfer (LODT) was the first cryptographic task with classical inputs and outputs proven to be unconditionally secure based on both quantum theory and special relativity. For the sake of clarity, throughout this section, we focus on the case where the parties agree on just two spacetime points. However, as noted in the original work [73], the LODT protocol can be easily extended to an arbitrarily higher number of spacetime points.

**Definition.** In LODT, both parties agree on two spacetime points, $Q_{0}$ and $Q_{1}$, and the receiver defines some $Q_{2}\in L\left(Q_{0}\right)\cap L\left(Q_{1}\right)$, where L(X) denotes the future light cone of spacetime point *X*. The sender inputs *just one* message *m*, and the receiver has no input. At the end of the protocol, the receiver obtains the message *m* at some random location $Q_{b}$ for b=0,1,2, while the sender stays oblivious about the spacetime point $Q_{b}$. Note that this is fundamentally different from SCOT. In SCOT, the receiver wants to hide his bit choice *b*, whether in LODT he wants to hide the *location* where he obtains the sender’s message *m*.

**Protocol [73].** The $\Pi^{\mathrm{LODT}}$ protocol assumes the sender and the receiver can independently and securely access all the points *P*, $Q_{0}$, $Q_{1}$ and $Q_{2}$, and instantaneously exchange information there. Theoretically, we achieve this through the concept of representatives (or agents) located at the relevant space-time points (*P*, $Q_{0}$, $Q_{1}$ and $Q_{2}$). Although the author [73] does not differentiate between agents, for the sake of coherence with SCOT exposition, here we simplify and refer to the sender’s agents as $S_{0}$, S and $S_{1}$ and to the receiver’s agents as $R_{0}$, R and $R_{1}$, which are located at x=−h, x=0 and x=h, respectively (Figure 9). Moreover, in the beginning of the protocol, the parties agree on a maximally entangled orthonormal basis of $H_{d}^{S}⊗H_{d}^{R}$ that encodes the possible messages owned by the sender, i.e., $\psi_{SR}^{i}$ for $i=1,\ldots ,d^{2}$. $H_{d}^{S}$ ($H_{d}^{R}$) is the *d*–dimensional Hilbert space initially used by the sender (receiver).

The protocol goes as follows. Instead of preparing a string of qubits based on conjugate coding, the agent S prepares a maximally entangled state encoding her message $m\in [d^{2}]$, i.e., $\psi_{SR}^{m}$. At point *P*, she sends the second subsystem $\psi_{R}^{m}$ to R. Then, each party choose randomly to which point ($Q_{0}$ or $Q_{1}$) they send their subsystem. If they happen to choose the same point $Q_{j}$, the agent $R_{j}$ is able to obtain message *m* at that point, for j=0,1. Otherwise, both receiver’s agents $R_{0}$ and $R_{1}$ have to send the corresponding subsystems $\psi_{S}^{i}$ and $\psi_{R}^{i}$ to some point $Q_{2}$ defined by the receiver. Since we are bounded by the laws of special relativity, the defined point $Q_{2}$ must be accessible from both $Q_{0}$ and $Q_{1}$. In other words, $Q_{2}$ must be in the intersection of their future light cones, i.e., $Q_{2}\in L\left(Q_{0}\right)\cap L\left(Q_{1}\right)$. Then, the receiver agent at $Q_{2}$ is able to make a joint measurement and obtains the integer *m*. Check the protocol in Figure 10 for more details.

**Figure 9 entropy-24-00945-f009:**
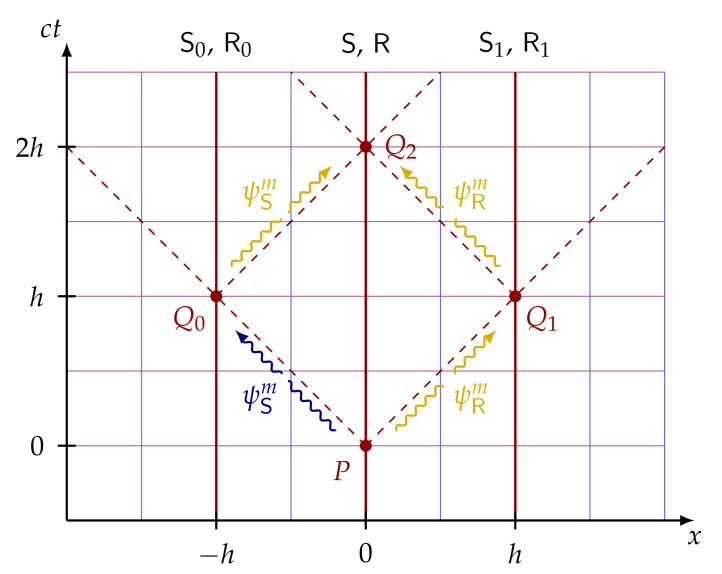
Representation of the $\Pi^{\mathrm{LODT}}$ protocol in the reference frame F in Minkowski spacetime where the sender randomly chooses j=0 and the receiver randomly chooses k=1. In this scenario, the receiver is only able to obtain message *m* at point *Q*. Blue arrows represent the information sent by the sender’s agents. Yellow arrows represent the information sent by the receiver’s agents.

## 5. Weak OT

In Section 3, we drew two research paths about quantum OT protocols that try to mitigate the impact of the impossibility results on the field of two-party quantum cryptography. In the previous section, we saw how the research community developed protocols based on some additional assumptions. In this section, we review some of the most important protocols that relax the definition of quantum OT, which we refer to as Weak OT (WOT). Similarly to the definition put forward by He [136], in WOT, both the sender and the receiver are allowed to cheat with some fixed probability. In other words, the sender has a specific strategy that allows her to find the receiver’s bit choice *b* with probability $p_{S}^{⋆}$, and the receiver has some strategy that allows him to obtain both messages $m_{0}$ and $m_{1}$ with probability $p_{R}^{⋆}$. The values $p_{S}^{⋆}$ and $p_{R}^{⋆}$ are commonly referred to as cheating probabilities and, ideally, should be strictly less than 1. The main aim of this line of research is to understand the physical limits of important cryptographic primitives based on protocols with no additional assumptions other than those imposed by the laws of quantum mechanics [136,137,138]. Consequently, these protocols “may not be well-suited for practical cryptography”, as stated by Chailloux et al. [137].

In this section, the two presented protocols are random OT. The sender does not define her messages, $m_{0}$ and $m_{1}$, and the receiver does not input his bit choice, *b*. Instead, they receive these elements as outputs. This feature is not a limitation of Weak OT protocols because “chosen” OT protocols can be reduced to random OT versions based on one-time-pad encryption [139].

**On bounds.** We already know that it is impossible to have an unconditionally secure QOT. However, the literature about WOT thrives to have a deeper understanding of these impossibility results by studying both upper and lower bounds on the cheating probabilities, $p_{S}^{⋆}$ and $p_{R}^{⋆}$. The Holy Grail of this research endeavour is to find protocols where both bounds meet, i.e., optimal protocols with tight cheating probabilities. The same endeavour was carried out successfully for quantum bit commitments [3] and quantum coin flipping [4]. However, at the time of writing, there has not been proposed an optimal protocol with tight cheating probabilities for OT under malicious adversaries. At present, only Chailloux et al. [138] presented a protocol that achieves the lower-bound cheating probability. However, it assumes the parties are semi-honest, i.e., the cheating parties do not deviate from the prescribed protocol.

The study of bounds on the cheating probabilities has two different approaches. On the one hand, more theoretical and non-constructive work has been done to find universal lower bounds, i.e., lower bounds on all possible QOT protocols. On the other hand, the search for stronger upper bounds follows a protocol-based approach, where each cheating probability is studied.

**On lower bounds.** It is common to look for the maximum value of the cheating probabilities when studying lower bounds. This is motivated by the fact that it is possible to develop a QOT protocol unconditionally secure against the sender ($p_{S}^{⋆}=\frac{1}{2}$) and completely insecure against the receiver ($p_{R}^{⋆}=1$) [6,47]. Therefore, the research community has been focused on finding general lower bounds on the maximum of the cheating probabilities, i.e., $p_{\max}^{⋆}:=\max\left(p_{S}^{⋆},p_{R}^{⋆}\right)$. At the time of writing, the known general lower bounds are presented in Table 1.

Next, we present two protocols proposed by the works [137,142] achieving a cheating probability $p_{\max}^{⋆}$ of 0.75.

**Chailloux-Kerenidis-Sikora protocol [137].** The first WOT protocol $\Pi_{\mathrm{wot}}^{\mathrm{CKS}}$ (Figure 11) was presented in a joint work by Chailloux, Kerenidis and Sikora [137]. This protocol is structurally different from BBCS-inspired protocols because it is a two-quantum-message protocol, i.e., the receiver sends some quantum system to the sender, and the sender returns the same quantum system to the receiver after applying some operation. Additionally, both parties work in a three-dimensional Hilbert space and do not use the standard conjugate coding technique. It is proved in the original work that both cheating probabilities are equal to 0.75, i.e., $p_{S}^{⋆}=p_{R}^{⋆}=0.75$.

The protocol is described in Figure 11 and goes as follows. The receiver starts by preparing an entangled state $|\phi_{b}\rangle$ that depends on his random bit choice *b*. Consequently, he saves one of the qutrits to himself and sends the other to the sender. After receiving the subsystem from the receiver, the sender applies a unitary operation according to her chosen random bit messages $m_{0}$ and $m_{1}$, and sends her subsystem back to the receiver. At this point in the protocol, the receiver owns a state $|\psi_{b}\rangle$ that is either orthogonal to the initial entangled state $|\phi_{b}\rangle$ or the same. Therefore, he can perform a measurement to perfectly distinguish these two cases. Since the message $m_{b}$ is encoded in the phase of the state $|\phi_{b}\rangle$, the receiver can conclude that $m_{b}=0$ when he obtains the initial state (i.e., no phase change) and $m_{b}=1$ when he obtains the corresponding orthogonal state $|\phi_{b}^{\prime }\rangle =\frac{1}{\sqrt{2}}\left(|bb\rangle -|22\rangle \right)$ (i.e., a phase change was applied).

**Figure 11 entropy-24-00945-f011:**
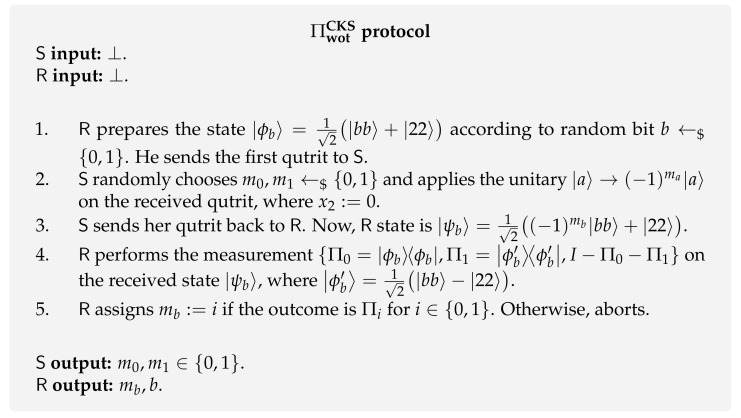
WOT protocol by Chailloux et al. [137].

**Amiri at al. protocol [142].** More recently, Amiri et al. [142] proposed a protocol $\Pi_{\mathrm{wot}}^{\mathrm{ASR}+}$ along with its experimental realization, that allows performing a batch of random WOT. The central technique used in this protocol is unambiguous state elimination (USE) measurements. Succinctly, unambiguous measurements aim to unambiguously distinguish a set of states $\rho^{x}$ for x∈X with prior probabilities $p_{x}$. USE measurements are a particular type of unambiguous measurements that only guarantee some state parameter *x* does not belong to a subset Y of X. In other words, these measurements decrease the set of possible states to which the measured state belongs. This protocol improves on the previous presented protocol $\Pi_{\mathrm{wot}}^{\mathrm{ASR}+}$, as the receiver’s cheating probability is slightly decreased to $p_{R}^{⋆}=0.73$.

The protocol is described in Figure 12 and goes as follows. In the first phase of the protocol, the sender starts by preparing a string of pairs of qubits of the form ${|x_{i}x_{i}\rangle }_{\theta_{i}}$, where $x_{i}\in \left\{0,1\right\}$ and $\theta_{i}\in \left\{+,\times \right\}$. This string of qubits encodes the random elements $m_{0}^{i}m_{1}^{i}\leftarrow_{\$}\left\{00,01,10,11\right\}$ generated by the sender that will lead to the final messages $m_{0},\;m_{1}\in {\left\{0,1\right\}}^{n-\sqrt{n}}$. The encoding is presented in the first step of the protocol $\Pi_{\mathrm{wot}}^{\mathrm{ASR}+}$. Note that, for each qubit *i*, the encoding mapping is designed in such a way that both the elements $m_{0}^{i}m_{1}^{i}$ encoded in the same basis $\theta_{i}$ and the corresponding encodings $|{x_{i}x_{i}\rangle }_{\theta_{i}}$ have opposite bits, i.e.,
$$\begin{array}{c} 00\to {|00\rangle }_{+}\;\;01\to {|00\rangle }_{\times }\\ 11\to {|11\rangle }_{+}\;\;10\to {|11\rangle }_{\times }. \end{array}$$
This separation is the key ingredient that allows a USE measurement to be carried out. After sending this string of qubits to the receiver, both parties engage in a *cut and choose phase*, where the receiver checks a subset of qubits, giving him confidence in the sender’s honesty. In the last phase, for each pair of qubits, the receiver performs one USE measurement on each qubit belonging to it. The USE measurements simply consist in measuring each qubit on a different basis. This will allow him to discard one element from the set of strings encoded by the computational basis, $Y_{+}=\left\{00,11\right\}$, and from the set of strings encoded by the Hadamard basis $Y_{\times }=\left\{01,10\right\}$. He will discard the elements by comparing the quantum state obtained in his measurements with the quantum states encoded in the corresponding basis. Now, the receiver takes as his message $m_{b_{i}}^{i}$ the bit that the remaining elements from both $Y_{+}$ and $Y_{\times }$ have in common and the choice bit $b_{i}$ the corresponding index.

As an example, imagine the sender uses the encoding of 00 to prepare the pair of qubits $00_{+}$ in round *i*. When measuring the first qubit on the computational basis, the receiver obtains $y_{0}^{i}=0$. In addition, he obtains randomly some $y_{1}^{i}$ when measuring the second qubit in the Hadamard basis. For the sake of exposition, let the element be $y_{1}^{i}=1$. Then, he discards the element 11 (encoded as $11_{+}$) from $Y_{+}$ because the state $0_{+}$ was obtained when the first qubit was measured on the computational basis. Similarly, he discards the element 01 (encoded as ${|00\rangle }_{\times }$) from $Y_{\times }$ because the state ${|1\rangle }_{\times }$ was obtained when measuring the second qubit in the Hadamard basis. The remaining strings are $y_{+,0}^{i}y_{+,1}^{i}=00$ and $y_{\times ,0}^{i}y_{\times ,1}^{i}=10$. By comparing both elements, the receiver outputs $m_{b_{i}}^{i}=0$ and $b_{i}=1$.

## 6. Weak Private Database Query

The concept of private database query (PDQ) was introduced by Gertner et al. [143] under a different name (private information retrieval), which is very similar to 1-out-of-*N* OT. The name is directly influenced by the following use case. One user is allowed to query just one database item without letting the owner of the database know which item was queried. The first quantum protocol for PDQ (also known as quantum database query) was proposed by Giovannetti et al. [144] and followed by additional works [145,146]. However, these protocols were not experimentally driven, and their implementation is rather difficult. The first experimentally feasible protocol was presented by Jakobi et al. [80].

In this section, we briefly review the work initiated by Jakobi et al. [80]. For the sake of consistency with previews sections, the user is called receiver (R) and the database owner is called sender (S). As this is a secure two-party quantum protocol, its security is affected by the aforementioned impossibility results [47]. Consequently, since Jakobi et al. protocol $\Pi^{\mathrm{PDQ}}$ (Figure 13 and Figure 14) is not based on any assumption model, the definition of PDQ has to be relaxed in order to allow its realization. Therefore, PDQ protocols fall into the category of 1-out-of-*N* Weak OT and, for this reason, we call it Weak PDQ. This line of research follows a more pragmatic approach as it is mainly focused on developing protocols (Table 2). In fact, to the best of our knowledge, the work by Osborn and Sikora [140] is the only one that studies theoretical bounds on the cheating probabilities of both parties for general two-party secure function evaluation, including 1-out-of-*N* OT.

**Protocol [80].** The first presented PDQ protocol $\Pi^{\mathrm{PDQ}}$ (Figure 13 and Figure 14) is very similar in structure to the BBCS $\Pi^{\mathrm{BBCS}}$ protocol [6]. Indeed, it is a one-quantum-message protocol that generats an *oblivious key* used by the sender to encode her database and by the receiver to obtain the desired item. In PDQ, we use the same definition of oblivious key (Definition 4) as given in Section 4.1. Besides the similarities between $\Pi^{\mathrm{PDQ}}$ and $\Pi^{\mathrm{BBCS}}$, the following differences are worth stressing.

Although the BBCS $\Pi^{\mathrm{BBCS}}$ protocol is insecure for a dishonest receiver, the $\Pi^{\mathrm{PDQ}}$ protocol guarantees that he only has a limited possibility of cheating. This improvement comes from the fact that $\Pi^{\mathrm{PDQ}}$ is based on the SARG04 Quantum Key Distribution (QKD) protocol [147] instead of the standard BB84 QKD protocol, which resists memory attacks to some extent. In fact, in the SARG04 protocol, the sender’s bases are never revealed to the receiver. Consequently, if the receiver postpones the measurement of the states, he is faced with a quantum discrimination problem, preventing him from having full knowledge of the photons’ state. Another distinctive feature of the SARG04 protocol is that it uses a modified version of quantum conjugate coding: BB84 states encode the key bits on the bases θ instead of encoding them on the vector elements x. This approach is adopted by Jakobi et al. [80] for the case of PDQ.

The full protocol is presented in both Figure 13 and Figure 14. It goes as follows. Similarly to the BBCS $\Pi^{\mathrm{BBCS}}$ protocol, the sender randomly prepares a string of qubits in randomly chosen bases, and the receiver measures the received qubits in random bases. Then, instead of revealing the sender’s bases $\theta^{S}$, for each index *i* the sender reveals a pair of states $\left\{{|a_{i}\rangle }_{u_{i}},{|b_{i}\rangle }_{v_{i}}\right\}$ drawn from four possibilities. Her choice is designed in such a way that one of the states in the pair is the one sent by her, and the other is in a random element on a different basis. Then, both parties are in a position to define their part of the shared oblivious key. The sender defines her oblivious key ${\mathrm{ok}}^{S}$ as the bases choices $\theta^{S}$ and the receiver defines ${\mathrm{ok}}^{R}$ based on the information given by the pair $\left\{{|a_{i}\rangle }_{u_{i}},{|b_{i}\rangle }_{v_{i}}\right\}$ and his measurements. At this stage, the receiver has around 1/4 of the elements of his oblivious key ${\mathrm{ok}}^{R}$ correlated with the sender’s oblivious key ${\mathrm{ok}}^{S}$. However, in PDQ, the receiver can only obtain one bit from the database. As such, they initiate a classical post-processing method that aims to reduce the receiver’s knowledge of the sender’s oblivious key ${\mathrm{ok}}^{S}$ to approximately one bit. Finally, the receiver tells the sender the required shift to be applied to the database, allowing him to decode the wanted database element through his oblivious key.

**Further work.** The above protocol $\Pi^{\mathrm{PDQ}}$ inspired the development of more efficient and flexible protocols for PDQ. In Table 2, we present a list of PDQ/OT protocols based on oblivious keys. Note that the term oblivious transfer (OT) is equivalent to private database query (PDQ), and QKD-based PDQ is equivalent to QOK-based OT. In addition, most of the protocols presented in Table 2 rely their security on the SARG04 protocol.

**Table 2 entropy-24-00945-t002:** Summary of PDQ research line.

Year	Author	Brief Description
2012	Gao et al. [148]	Generalized the $\Pi^{\mathrm{PDQ}}$ [80] protocol by adding a parameter θ that regulates the average number of bits known by the receiver.
2013	Rao et al. [149]	Improved the communication complexity of $\Pi^{\mathrm{PDQ}}$ [80] from O(NlogN) to O(N).
2013	Zhang et al. [150]	Designed a PDQ protocol based on counterfactual QKD.
2014	Wei et al. [151]	Developed a generalization of the $\Pi^{\mathrm{PDQ}}$ [80] protocol that allows to retrieve a block of bits from the database with only one query.
2014	Chan et al. [152]	Developed a practical fault-tolerant PDQ protocol that can cope with noisy channels and presented an experimental realization.
2015	Gao et al. [153]	Presented an attack on the common dilution method of the oblivious key and introduced a new error-correction method for the oblivious keys.
2015	Liu et al. [154]	Introduced a PDQ protocol based on Round Robin Differential Phase Shift (RRDPS) QKD which limits the number of items an honest receiver is able to know to just one and with zero failure probability.
2015	Yang et al. [155]	Proposed the first PDQ protocol based on semi-QKD.
2015	Yu et al. [156]	Pointed that the Yang et al. [155] semi-QKD based PDQ protocol can be attacked and presented a fully quantum PDQ.
2016	Wei et al. [157]	Proposed a two-way QKD based PDQ protocol that is loss tolerant and robust against both quantum memory and joint measurement attacks.
2016	Yang et al. [158]	Proposed a PDQ protocol based on one-way-six-state QKD with security against joint-measurement attacks given by a new design for the classical post-processing of the oblivious keys.
2017	Maitra et al. [159]	Proposed a Device-Independent Quantum Private Query.
2018	Wei et al. [160]	Examined the security of Liu et al. [154] RRDPS protocol under imperfect sources and presented an improved protocol based on a technique known as low-shift and addition (LSA).
2018	Zhou et al. [161]	Proposed a new PDQ protocol based on two-way QKD that ensures the privacy of both sender and receiver.
2019	Chang et al. [162]	Suggested a PDQ protocol based on a two-way QKD with improved privacy.
2019	Du and Li [163]	Proposed a robust High Capability QKD-Based PDQ protocol.
2020	Ye et al. [164]	Developed a Semi-QKD based PDQ protocol such that any kind of evasion can be detected.

## 7. Further Topics

The research field of quantum oblivious transfer is already quite extensive and, in this review, but we only focus this review on a particular type of OT, namely 1-out-of-*N* OT. We briefly mention some topics that can be included in an extended version of this work.

**All-or-nothing OT.** The first proposal of OT was put forward by Rabin [10] in a flavour different from 1-out-of-2 OT, named *all-or-nothing* OT or 1/2 OT. In this flavour, the sender only has one message *m*, and the receiver receives it with probability 1/2, without the sender knowing whether or not the receiver has received her message. In the classical setting, both 1-out-of-2 OT and all-or-nothing OT are proved to be equivalent [165]. However, these classical reductions cannot be applied in the quantum setting as it was proved by He and Wang [166] that these two flavours are not equivalent in the quantum setting. The first all-or-nothing QOT was proposed by Crépeau and Kilian [82] and later extended by Damgård et al. [68] in the bounded-quantum-storage model. In general, 1-out-of-2 OT protocols can be adapted to achieve all-or-nothing OT [167,168]. Moreover, He and Wang [169] presented an entanglement-based all-or-nothing OT protocol that claims to be secure despite the impossibility results of two-party function evaluation. Their claim is based on the fact that, in the all-or-nothing variant, the receiver does not unambiguously obtain the message *m*, which is an implicit assumption in Lo’s impossibility result [47].

**XOR OT.** The concept of XOR oblivious transfer was presented in the classical setting by Brassard et al. [170]. In this variant of OT, the sender inputs two messages, $m_{0}$ and $m_{1}$, and the receiver obtains one of these three elements: $m_{0}$, $m_{1}$ or $m_{2}=m_{0}⊗m_{1}$. In the quantum setting, there are currently only two proposed protocols that achieve this task [171,172].

**OT of qubits.** The vast majority of quantum oblivious transfer protocols focus on a classical input-output setting, i.e., both the messages input by the sender and the elements obtained by the receiver are classical. More recently, Zhang et al. [173] proposed the concept of OT with qubit messages. In their work, they present a variant of the all-or-nothing OT with an unknown qubit message. The main tool used to achieve this task is a probabilistic teleportation protocol.

**Experimental protocols.** Experimental realizations of quantum communication protocols have to take into account sources of errors (loss of photons and error in measurements) which are not considered by more theoretical protocols. In practice, it is desirable to design loss-tolerant and fault-tolerant protocols. This study was initiated by Schaffner et al. [105,109] and followed by Wehner et al. [174], where they analyse the impact of both loss and error on the security of the protocol. Based on this work, two independent practical experiments implemented OT in the noisy storage model. Erven et al. [175] implementation was based on Discrete Variables and generated a 1366-bit random oblivious transfer string in ∼3 min. Furrer et al. [176] implementation was based on Continuous Variables and achieved a generation rate of around 1000 oblivious bit transfers per second. In addition, experimental implementations of PDQ protocols have been reported in the literature [152] as well as Weak OT protocols [142].

## 8. Conclusions

Since the first proposal of quantum OT 40 years ago, active and fruitful research around this topic deepened our understanding of the limits and advantages of quantum cryptography. It was first proved that two fundamental primitives, bit commitment and oblivious transfer, are equivalent in the quantum setting, a relation that does not hold classically. Unfortunately, it was also proved that both primitives cannot be unconditionally secure in the quantum setting, matching the impossibility results in the classical setting. However, this equivalence in the quantum setting implies that quantum OT requires weaker security assumptions than classical OT. Quantum OT can be implemented solely with quantum-hard one-way functions and classical OT requires at least one-way functions with trapdoors, i.e., some sort of asymmetric cryptography. This makes classical OT potentially more vulnerable to quantum computer attacks and tendentiously less computationally efficient. Additionally, some quantum OT implementations benefit from an important feature, known as everlasting security, that does not have a classical counterpart. It states that even if the security assumptions lose validity after the protocol execution, the security of the protocol is not compromised. In other words, quantum OT implementations are considered unconditionally secure after the protocol execution.

We went through some of the most common assumptions used to implement secure quantum OT. Hybrid approaches are based on both quantum physical laws and computational complexity assumptions. These can offer practical and secure solutions, with gains both in terms of security and efficiency, when compared with classical implementations. Limited-quantum-storage approaches offer secure solutions as long as the technological limitations are met during the protocol execution. In addition, two primitives inspired by OT are shown to be unconditionally secure under relativistic effects. Interestingly, these are not possible in the classical setting. Protocols solely based on the laws of quantum mechanics lead to protocols where the parties can cheat with some fixed probability. These protocols are commonly explored in the subfields of weak OT and private database query.

## Figures and Tables

**Figure 1 entropy-24-00945-f001:**
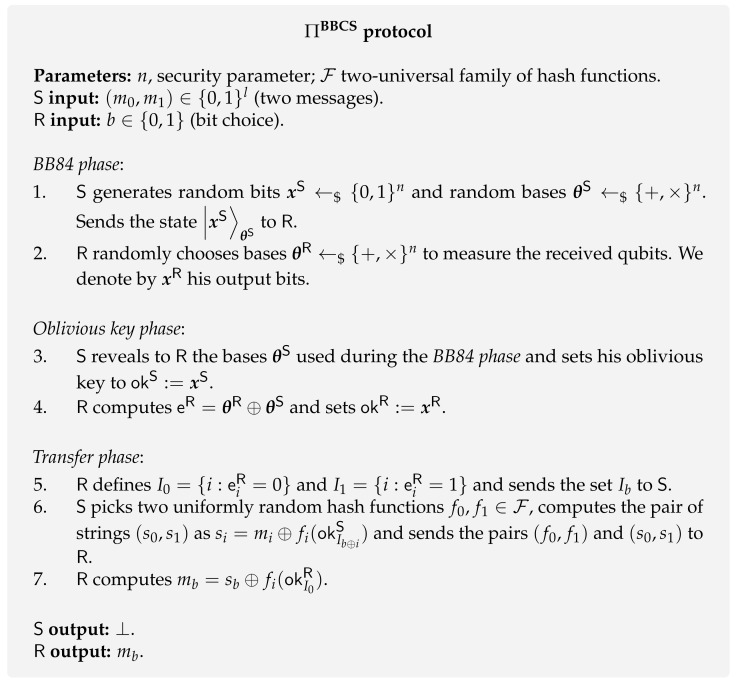
BBCS OT protocol.

**Figure 2 entropy-24-00945-f002:**
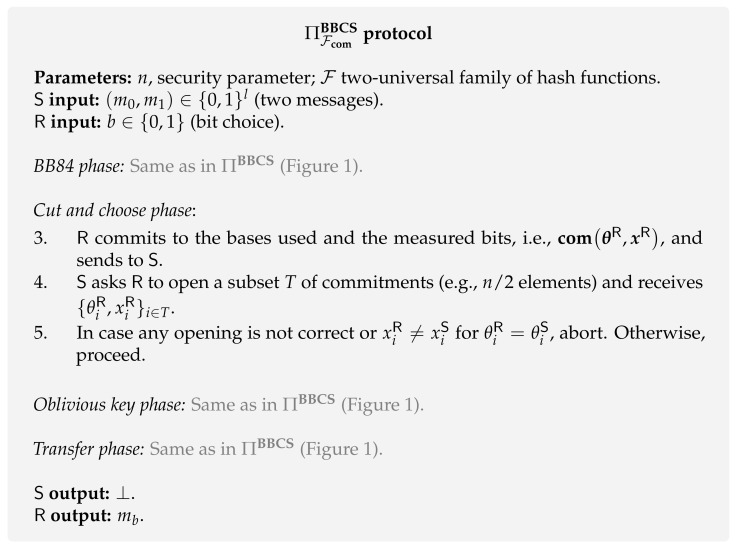
BBCS OT protocol in the $F_{\mathrm{com}}$–hybrid model.

**Figure 3 entropy-24-00945-f003:**
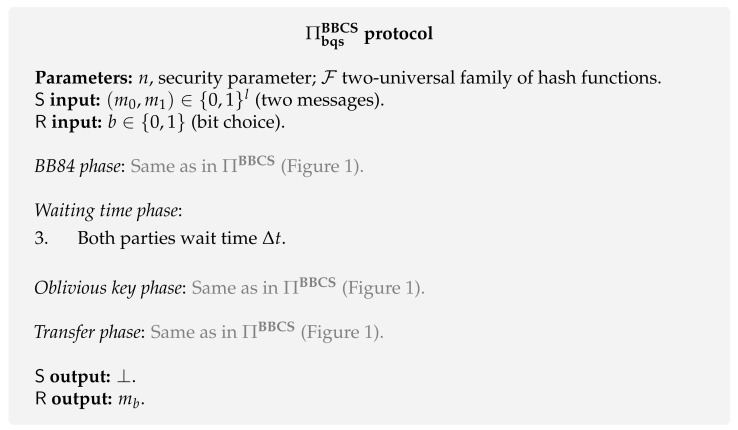
BBCS OT protocol in the bounded-quantum-storage model.

**Figure 4 entropy-24-00945-f004:**
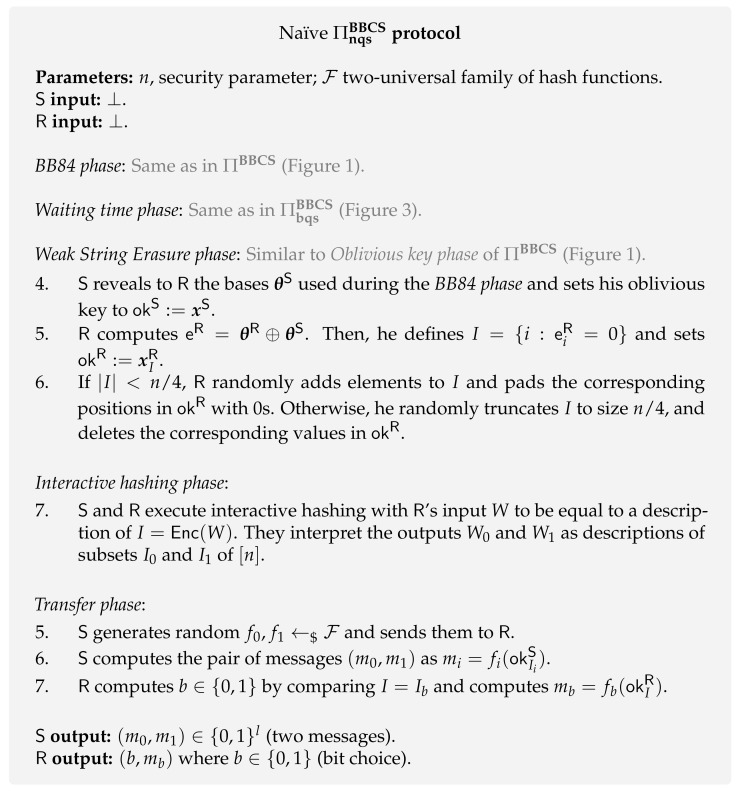
BBCS OT protocol in the noisy-quantum-storage model.

**Figure 6 entropy-24-00945-f006:**
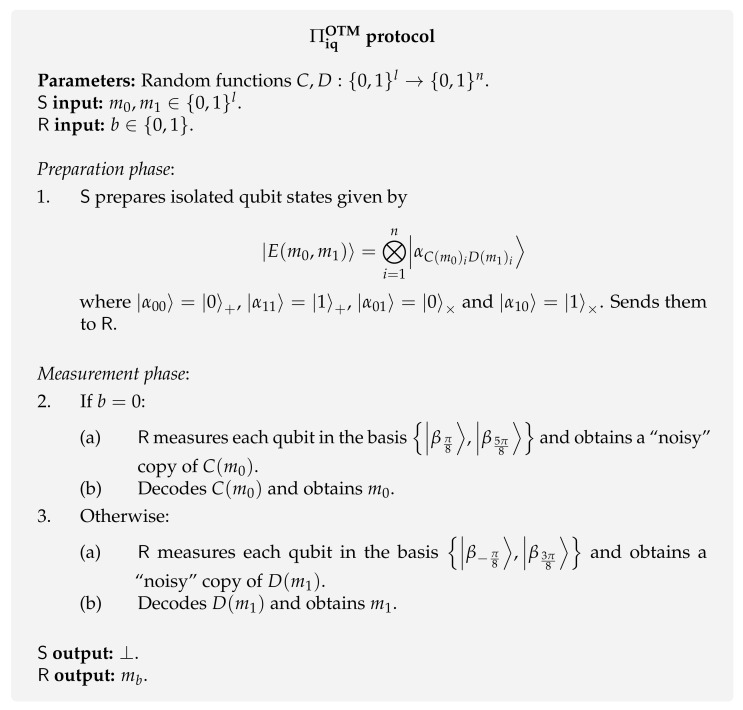
OTM protocol in the isolated-qubits model [71].

**Figure 7 entropy-24-00945-f007:**
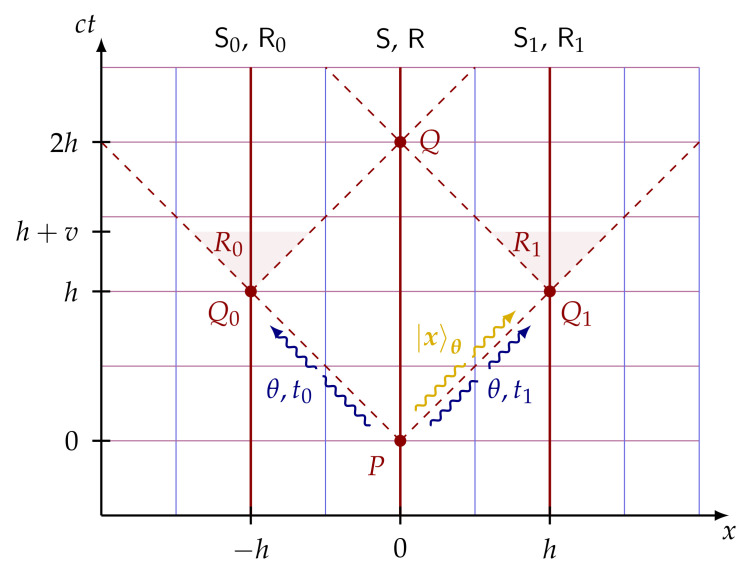
Representation of the $\Pi^{\mathrm{SCOT}}$ protocol in the reference frame F in Minkowski spacetime where the receiver chooses b=1. In this scenario, the receiver obtains message $m_{1}$ at point $Q_{1}$. Note that the receiver can retrieve the message $m_{0}$ at point *Q*. This event does not compromise the SCOT security definition because it only demands that $m_{0}$ is not known at space-time region $R_{0}$. More specifically, at point *Q*, the receiver can use the key *x* to compute $m_{0}$ from the encrypted value $t_{0}$ he received at point $Q_{0}$. Blue arrows represent the information sent by the sender’s agents. Yellow arrows represent the information sent by the receiver’s agents. Adapted from the original article [72].

**Figure 8 entropy-24-00945-f008:**
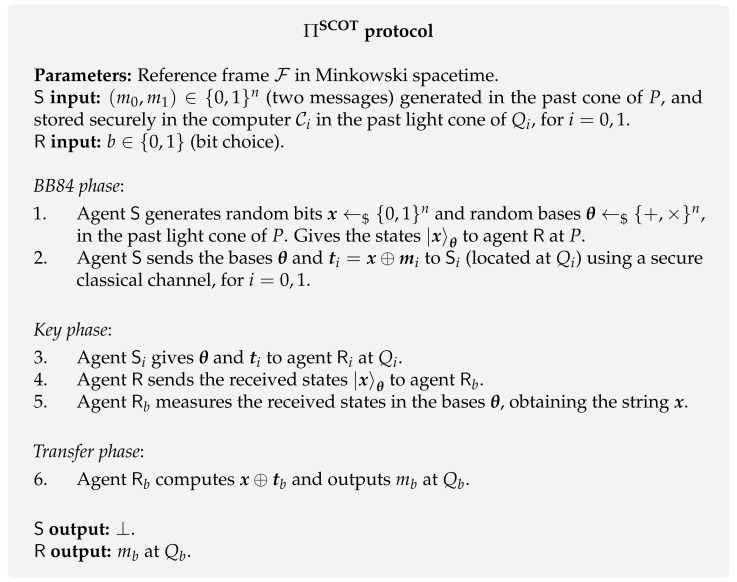
SCOT protocol [72].

**Figure 10 entropy-24-00945-f010:**
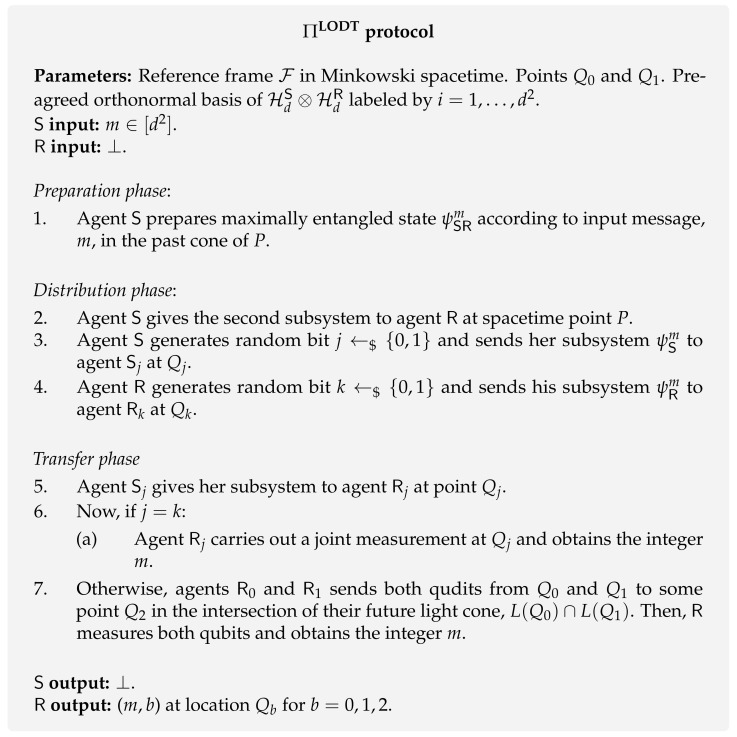
LODT protocol [73].

**Figure 12 entropy-24-00945-f012:**
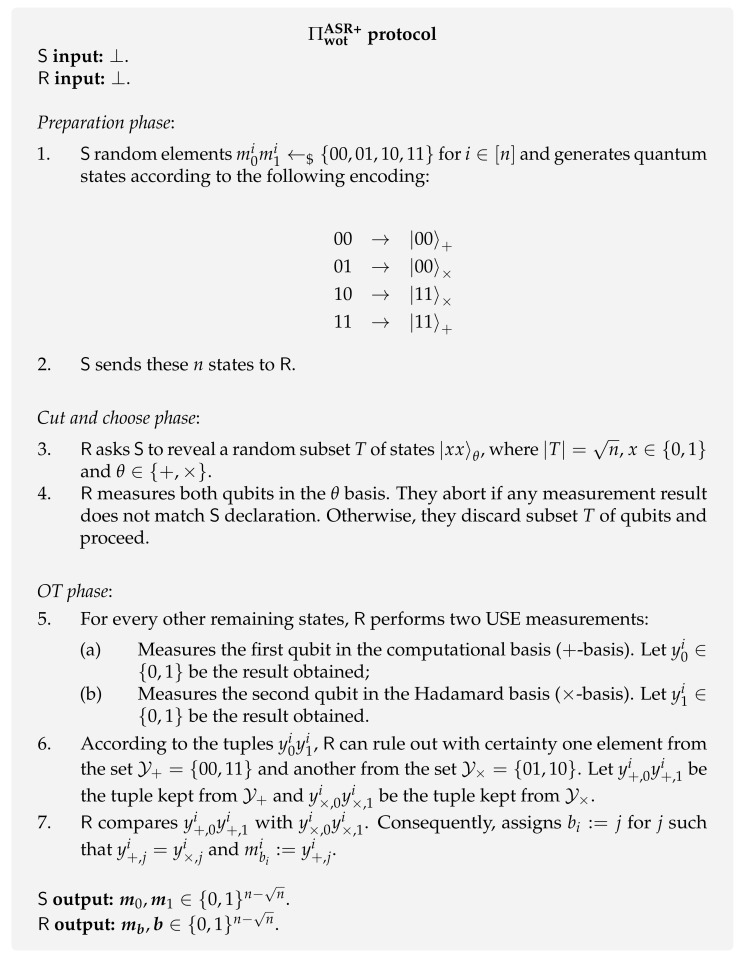
WOT protocol by Amiri et al. [142].

**Figure 13 entropy-24-00945-f013:**
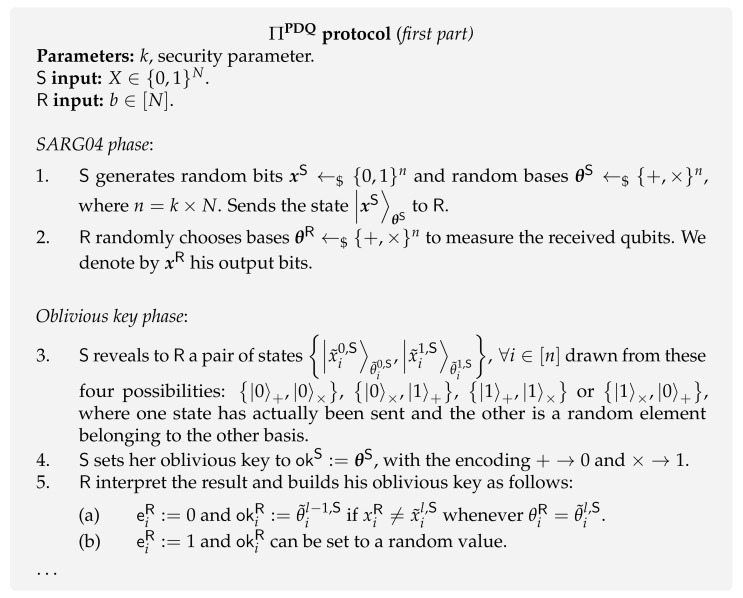
First part of the PDQ protocol by Jakobi et al. [80].

**Figure 14 entropy-24-00945-f014:**
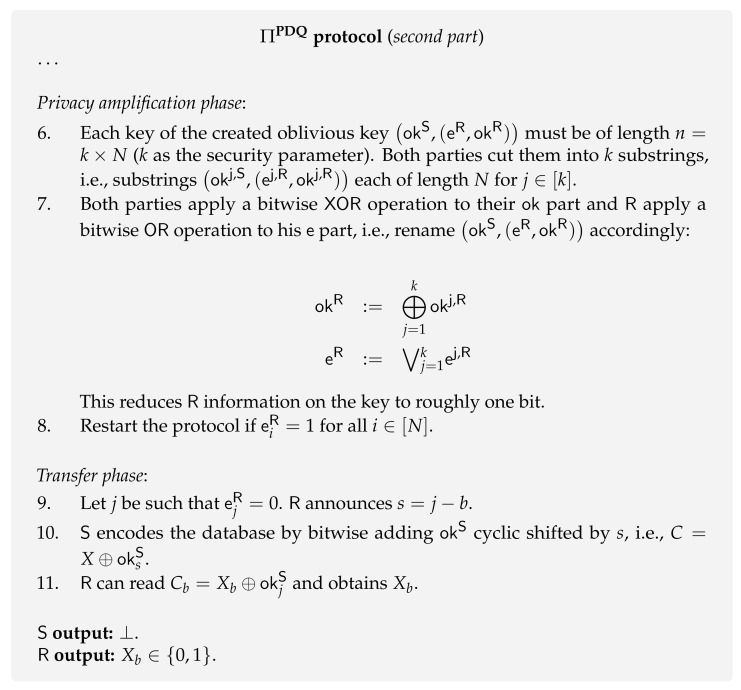
Second part of the PDQ protocol by Jakobi et al. [80].

**Table 1 entropy-24-00945-t001:** General lower bounds on $p_{\max}^{⋆}$.

Ref.	[140]	[137]	[141]	[138] ^1^ [142]
$p_{\max}^{⋆}\geq$	0.52	0.59	0.61	0.67

^1^ In this work, the authors restrict the analysis to semi-honest QOT protocols.

## Data Availability

Not applicable.

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
