# Peer review of "Quantum Oblivious Transfer: A Short Review"

_entropy, 2022, doi:10.3390/e24070945_

Round 1
Reviewer 1 Report
In the article, the authors survey the work developed around the concept of oblivious transfer in the area of theoretical quantum cryptography, with an emphasis on some proposed protocols and their security requirements. The authors went through some of the most common assumptions used to implement secure quantum OT.
Existing protocols are newly defined and organized, and various properties are well mentioned. I agreed that hybrid approaches based on quantum physical laws and computationally complexity assumptions can offer a practical and secure solution, with gains both in terms of security and efficiency when compared with classical implementations.
The presented article seem to be meaningful and valid enough to be published. Also, the form of the manuscript meets the standards of the Entropy journal. Although it is written in a highly technical manner, the article could be interesting for readers who are not necessarily experts in the field. In my opinion, the submitted article can be published after fixing some minor typos.
Author Response
Point 1. The submitted article can be published after fixing some minor typos.
Ans: Thank you! We went through the paper and edited the English language and the typos.
Reviewer 2 Report
The manuscript reviews an important field of quantum cryptography called the quantum oblivious transfer (quantum OT, QOT). The main part of quantum cryptography is quantum key distribution (QKD), but there are also other important cryptographic tasks solved by means of quantum mechanics. These branches of quantum cryptography are less known. Typically, general reviews on quantum cryptography mainly review QKD and only briefly review other tasks. Thus, good and informative reviews devoted to separate less known tasks are very desirable. So, I find the present manuscript very important.
This is a very good review. The authors give a very detailed introduction into the history of the subject and give an exhaustive number of references on both the notion of composable security in classical and quantum cryptography and on the discussed topic of QOT. In the description of protocols, I think, the authors have found an ideal balance: From one side, the descriptions of the protocols are rather informative; from the other side, the review is not overloaded by details. This makes the review interesting for a wide audience working in the field of quantum information. Also I find advantageous that the authors give both formal algorithmic descriptions of the protocols (as “figures”) and less formal explanations in the main text.
With pleasure, I recommend the paper for publication. However, I also recommend the authors to address the following issues in the manuscript:
1. Is the problem of OT practical or just theoretical and mathematical? If it is practical, then it is worthwhile to give a pair of concrete examples of practical application. Application to multipartite computations are mentioned, but it is desirable to give concrete examples where the task of OT is or can be applied. Do people use classical OT somewhere? E.g., in line 883, it is mentioned that one experimental scheme allows one to perform 1366 bit OT in 3 min. So, OT is interesting not just a rare procedure, but a frequent one?
2. Lines 137-139: The notion of two-sided two-party computation is unclear. If there is no restriction that only one side is allowed to learn the result, then it is simply the usual computation. How can it be impossible?
3. Line 163: From this phrase, one can understand that Private database query (PDQ) itself is a weak form of OT. However, as we read below in Sec. 6, PDQ is another name for 1-out-of-N OT and, so, originally is not weak. But if we do not impose any additional assumptions, then one can ensure only a weak form of PDQ. Probably, to avoid misunderstanding, it is better to replace “Weak OT (Section 5) and Private Database Query (Section 6)” by “Weak OT (Section 5) and Weak Private Database Query (Section 6)”.
4. Figure 3: What is \Delta t? This quantity is introduced later, in the noisy-quantum-storage model. However, originally, Figure 3 describes the protocol in the bounded-quantum-storage model, which does not contain this parameter.
5. Lines 448 and 453: What are Q, H_Q and H_{Q_out}?
6. Section 4.6.1. In Lines 599-601, it is written: “These two variants do not exactly follow the OT definition as it was proved that it is impossible to construct unconditionally secure OT even under the constraints imposed by special relativity”. However, why the SCOT (space-time constrained OT) protocol is not a fully satisfactory solution of the OT problem? The receiver gets the desired message in the corresponding region of space-time. Then both agents R_0 and R_1 can securely (using QKD and the one-time-pad encryption) the following two bits sent to the region R. If b=0 and R_0 has obtained the bit m_0, then he sends the bit 1 (which means that he has obtained a message) and m_0 to R, while R_1 sends the bit 0 (which means that he has not obtained a message) and a random bit to R. Then R knows exactly m_b, but not m_{1-b}, while the sender is still ignorant about b. Depending on the answer to this question, please, modify the corresponding sentence in Conclusion accordingly.
6. In Section 4, we consider the messages m_0 and m_1 and bit b as given inputs. In Section 5, they become random. Such modification should be explained. Is this statement also practical or we follow the statement of Line 696 that here we more investigate the capabilities of quantum information rather than develop protocols for practical cryptography. From the other side, again, using the one-time-pad encryption, one can easily turn a transfer of random messages into a transfer of concrete messages.
7. Line 669: It is said about “some” orthonormal bases. But later it is used that they are maximally entangled. If I understand correctly, “maximally entangled” should be added here or replace the word “some”.
8. In line 708, the notion of semi-honest was mentioned, but not explained.
9. The PDQ protocol, the second part, Fig. 14. Are the descriptions of step 7 and both step 8 (it seems that two steps 8 is a typo) with respect to the value of e correct? If I understand step 5 correctly, if e^R_i=0, then then the corresponding bits of ok^S and ok^R coincide, whereas if e^R_i=1, then the receiver has no information about sender’s bit. If so, it seems that, on step 7, the receiver should perform bitwise OR operation to his e part, restart the protocol whenever e^R_i=1 for all i, and continue if there is certain j such that e^R_j=0. The last condition means that there is at least one position j where the receiver knows that his bit of the oblivious key coincide with the corresponding sender’s bit.
Minor comments and typos:
1. Line 63: The explanation of the abbreviation BBCS should be given (since this is the first instance of this abbreviation). Also the explanation of BCJL on Line 128 should be given.
2. Line 99: “one messages” should be replaced by “one message”.
3. Line 116: “knwon” should be replaced by “known”.
4. Line 128: The sentence now says that “a protocol” presented “a flawed proof”. It seems that something is wrong: Certain authors (rather than a protocol) could presented one or another proof.
5. Lines 129, 130, and 141: “Mayer” should be replaced by “Mayers”.
6. Lines 156-159. Point 2 (Lines 160-163) has references to concrete sections in the paper: Sections 5 and 6. Probably, for consistency, it would be better to add a reference to Section 4 in point 1.
7. Line 191: Does H_2 denote a two-dimensional space? Also Line 669: Does the superscript d denotes the dimensionality?
8. Figure 1. It is better to specify the domain and the range of the hash functions F.
9. Figure 1. I cannot find an explanation of the notation "left arrow with $ subscript".
10. Line 403: Probably, a typo in the symbol \subset (\subset \in H_out).
11. Formula after Line 453: It should be mentioned that H with a superindex denotes the smooth min-entropy.
12. Line 516: “the fact that it not known”: Probably, “is” is missed.
13. Line 614: “time region”, probably, should be replaced by “space-time region”.
14. Line 732: It seems that “p_R^*=p_R^*=0.75” should be replaced by “p_S^*=p_R^*=0.75”.
15. Figure 13, step 3: I think, it would be better to stress the randomness of the choice of a state from the other basis.
16. Figure 14, step 9: I think, it would be better to add “cyclic” before “shift”.
Reviewer 3 Report
In this manuscript, the authors focus the quantum oblivious transfer. They provide the comparison between the application structure of both QKD and QOT primitives. It is shown that QKD allows quantum –safe communication. Actually, besides the QKD, there is another important safe communication. QKD is to share the key. Quantum secure direct communication (QSDC) is another quantum communication branch which can transmit secret message without sharing a key. In this review, it is also better to add some discussion about the QSDC, I think it will be helpful for this review. Some related work about QSDC are suggested.
[1] Chen SS, Zhou L, Zhong W, et al. Three-step three-party quantum secure direct communication. Sci China Phys Mech Astron 2018;61:090312.
[2] Li T, Long GL. Quantum secure direct communication based on single-photon bell-state measurement. New J Phys 2020;22:063017
[3]Y. B. Sheng, L. Zhou, and G. L. Long, One-step quantum secure direct communication, Sci. Bull. 67, 367 (2022)
Author Response
Point 1. In this review, it is also better to add some discussion about the QSDC, I think it will be helpful for this review. Some related work about QSDC are suggested.
Thank you for suggesting the QSDC concept. We included it in the introduction along with the suggested references.
"Furthermore, besides QKD, there is another important safe communication primitive called quantum secure direct communication (QSDC) [10-13]. In this branch, one can transmit secret messages without sharing a key."
Point 2. Extensive editing of English language and style required
Thank you for the suggestion. We went through the article and thoroughly revised and edited the English language.